# MASK IN THE MIRROR: IMPLICIT SPARSIFICATION

**Tom Jacobs**
CISPA Helmholtz Center for Information Security
tom.jacobs@cispa.de

**Rebekka Burkholz**
CISPA Helmholtz Center for Information Security
burkholz@cispa.de

## ABSTRACT

Continuous sparsification strategies are among the most effective methods for reducing the inference costs and memory demands of large-scale neural networks. A key factor in their success is the implicit $L_1$ regularization induced by jointly learning both mask and weight variables, which has been shown experimentally to outperform explicit $L_1$ regularization. We provide a theoretical explanation for this observation by analyzing the learning dynamics, revealing that early continuous sparsification is governed by an implicit $L_2$ regularization that gradually transitions to an $L_1$ penalty over time. Leveraging this insight, we propose a method to dynamically control the strength of this implicit bias. Through an extension of the mirror flow framework, we establish convergence and optimality guarantees in the context of underdetermined linear regression. Our theoretical findings may be of independent interest, as we demonstrate how to enter the rich regime and show that the implicit bias can be controlled via a time-dependent Bregman potential. To validate these insights, we introduce PILoT, a continuous sparsification approach with novel initialization and dynamic regularization, which consistently outperforms baselines in standard experiments.

## 1 INTRODUCTION

Deep learning continues to impress across disciplines ranging from language and vision (Ramesh et al., 2022) to drug design (Stephenson et al., 2019; Jumper et al., 2021) and even fast matrix multiplication (Fawzi et al., 2022). These accomplishments come at immense costs, as they rely on increasingly large neural network models. Moreover, training such massive models with first-order methods like variants of Stochastic Gradient Descent (SGD) is a considerable challenge and often requires large-scale compute infrastructure (Kaack et al., 2022). Even higher costs are incurred at inference time, if the trained models are frequently evaluated (Wu et al., 2022; Luccioni et al., 2023).

Sparsifying such neural network models is thus a pressing objective. It not only holds the promise to save computational resources, it can also improve generalization (Frankle & Carbin, 2019; Paul et al., 2023), interpretability (Chen et al., 2022; Hossain et al., 2024), denoising (Jin et al., 2022; Wang et al., 2023), and verifiability (Narodytska et al., 2020; Albarghouthi, 2021). However, at its core is a hard large-scale nested optimization problem combining multiple objectives. In addition to minimizing a typical neural network loss $\min_{w \in \mathbb{R}^n} f(w)$ (and its generalization error), we wish to rely on the smallest possible number of weights, effectively minimizing the $L_0$ norm $\min_{w \in \mathbb{R}^n} ||w||_{L_0}$. This is an NP-hard problem that is also practically hard to solve due to its mixed discrete and continuous nature. This becomes more apparent when we reformulate it like the best performing sparsification methods.

Among such approaches that achieve high sparsity while maintaining high generalization performance are continuous sparsification methods and iterative pruning strategies. These methods explicitly identify for each weight parameter $w$ of a neural network a binary mask $m \in \{0, 1\}$ that signifies whether a parameter is pruned. Thus, a parameter is set to zero ($m = 0$) or not ($m = 1$), effectively parameterizing the network with parameters $x = m \odot w$, where $\odot$ is the pointwise multiplication (Hadamard product). The introduction of the additional mask parameters $m$ turns the sparsity objective into a discrete $L_1$ penalty of $m$ as $||w \odot m||_{L_0} = \sum_i m_i$ subject to $m_i \in \{0, 1\}$, where $\odot$ denotes elementwise multiplication and we assume that $w \neq 0$ if $m = 1$. The $L_1$ objective is already more amenable to continuous optimization than the original $L_0$ objective (Louizos et al., 2018). Nevertheless, the big challenge arises from the fact that $m$ is binary.

Continuous sparsification addresses this issue by relaxing the optimization problem to continuous or even differentiable variables $m$, often with $m \in [0, 1]$ by learning a parameterization $m = g(s)$ with $g: \mathbb{R} \to [0, 1]$, e.g., like a sigmoid. This way, the problem becomes solvable with standard first-order optimization methods. Yet, moving from the continuous space back to the discrete space is error-prone. Regularizing and projecting $m$ towards binary values $\{0, 1\}$ is generally problematic, requires careful tuning, and often entails robustness issues.

However, this projection step is not necessarily required in a parameterization $m \odot w$, where $m$ can freely attain values in $\mathbb{R}$. Ziyin & Wang (2023) show that a loss with this parameterization $m \odot w$ combined with weight decay $\alpha \left( ||m||_{L_2}^2 + ||w||_{L_2}^2 \right)$ is equivalent to LASSO and thus an explicit $L_1$-regularization (Ziyin, 2023). Yet, their proposed method spred significantly outperforms LASSO, which suggests that the posed equivalence of optimized objectives cannot explain this success.

As we show, an important, but overlooked, distinguishing factor is the fact that continuous sparsification with $m \odot w$ parameterization is driven by an implicit rather than an explicit regularization. This implies that in a sufficiently overparameterized setting, the following hierarchical optimization problem is solved:

$$\min_{x \in \mathbb{R}^n : f(x)=0} ||x||_{L_1}. \tag{1}$$

The main idea follows the same philosophy as the lottery ticket hypothesis (Frankle & Carbin, 2019). From a range of models which all attain optimal training loss, it prefers the sparsest model. In other words, we aim to find a subnetwork with a similar accuracy as a dense network. Instead of having two competing objectives, we enforce cooperation between the two objectives by subjugating the sparsification. While this formulation can have advantages, if we can attain zero training loss ($f(x) = 0$), it might still be equivalent to an explicit regularization.

The real potential of continuous sparsification and its induced implicit bias (in particular in non-convex settings) becomes apparent when we study the corresponding learning dynamics. Our theoretical analysis of (stochastic) gradient flow applied to $m \odot w$ reveals that the dynamics differ fundamentally from the ones of LASSO, where redundant features are sparsified exponentially fast. Instead, we show by extending the mirror flow framework that a dynamic explicit weight decay can move the implicit bias from $L_2$ to $L_1$ during training. In consequence, the sparsification becomes effective only relatively late during training, allowing the overparameterized model to first attain high generalization performance. The dynamics resemble thus a successful strategy that is applied across continuous and iterative pruning methods, which all acknowledge and realize the premise that training overparameterized models before they are sparsified usually leads to significant performance benefits (Frankle & Carbin, 2018; Paul et al., 2023; Gadhikar & Burkholz, 2024).

Our analysis extends the implicit bias framework, which covers different parameterizations (Pesme et al., 2021; Gunasekar et al., 2020; Woodworth et al., 2020; Li et al., 2022) and the well-known fact that standard gradient flow has an implicit $L_2$ bias (Nemirovski & Yudin, 1983; Beck & Teboulle, 2003). Another common use of the framework has been to study when training dynamics enter the so-called rich regime, which is responsible for improved feature learning. In this context, we study two main innovations. a) As we show, the explicit regularization (i.e. weight decay on $m$ and $w$) guides the strength of the implicit bias. The fact that this makes the implicit bias tuneable makes it practically relevant for sparsification, as we have to be able to reach a target sparsity. b) By proposing a dynamic regularization (rather than a common static one), we obtain control over the transition speed from $L_2$ to $L_1$ regularization, which is crucial for performance gains in the high sparsity regime and enables us to enter the rich regime. From a conceptual point of view, we unite explicit and implicit bias within a time-dependent Bregman potential, which is potentially of independent theoretical interest.

While our general derivations provide insights into various continuous sparsification approaches, including STR (Kusupati et al., 2020), spred Ziyin & Wang (2023), or (Savarese et al., 2021), we also utilize them to propose a new improved algorithm, PILoT (Parametric Implicit Lottery Ticket). PILoT combines the $m \odot w$ parameterization with a dynamic regularization and an initialization that enables sign flips. Such sign flips are key to effective sparse training (Gadhikar & Burkholz, 2024), but are not feasible with the spred initialization. The dynamic regularization and thus implicit bias leads us to outperform state-of-the-art baselines in particular in the high-sparsity regime, as we demonstrate in extensive experiments.

In summary, we make the following **contributions:**

- We gain novel insights into continuous sparsification by highlighting its implicit bias towards sparsity induced by doubling the number of trainable parameters. In particular, we explain the effectiveness of spred (Ziyin & Wang, 2023), which is based on $m \odot w$.

- To the best of our knowledge, we are the first to introduce the implicit bias with an explicit regularization resulting in a mirror flow with a time-dependent Bregman potential.

- We provide convergence results for (quasi)-convex loss functions (Theorem 2.2) and optimality for underdetermined linear regression (Theorem 2.3) with time-dependent Bregman potential.

- Improving results by (Alvarez et al., 2004; Li et al., 2022), we replace convexity with the Polyak-Łojasiewicz inequality, quasi-convexity and a growth condition on the Bregman potential (see Theorem A.3).

- Using our extensions of the mirror flow framework, we propose a new continuous sparsification method, PILoT, which controls the implicit regularization dynamically moving from $L_2$ to $L_1$. Its initialization enables sign flips in contrast to spred.

- In experiments for diagonal linear networks and vision benchmarks (including ImageNet), PILoT consistently outperforms baseline sparsification methods such as STR and spred, which demonstrates the utility of our theoretical insights.

## 1.1 RELATED WORK

**Neural network sparsification.** A multitude of neural network sparsification methods have been proposed with different objectives (Liu & Wang, 2023). A popular objective is, for instance, to save computational and memory costs primarily at inference, or also during training, which is linked to the time of pruning, i.e., initially (Frankle et al., 2021; Lee et al., 2019; Tanaka et al., 2020; Wang et al., 2020; Pham et al., 2023; Patil & Dovrolis, 2021; Tanaka et al., 2020; Liu et al., 2021; Gadhikar et al., 2023; Fischer & Burkholz, 2021), early during training (Evci et al., 2020; Dettmers & Zettlemoyer, 2019), during training like continuous sparsification (Sreenivasan et al., 2022; Kusupati et al., 2020; Savarese et al., 2021; Peste et al., 2021) or within multiple pruning-training iterations (Han et al., 2015; Frankle & Carbin, 2018; You et al., 2020; Renda et al., 2020; Gadhikar & Burkholz, 2024). Other distinguishing factors are which type of sparsity the methods seek, if they focus on saving computational resources and memory in specific resource-constrained environments or, which methodological approach they follow.

**Unstructured sparsity.** In this work, we focus on unstructured sparsity, i.e., the fraction of pruned weights, and thus seek to remove as many weight entries as possible, which can achieve generally the highest sparsity ratios while maintaining high generalization performance. Structured sparsity, which usually obtains higher computational gains on modern GPUs (Kuzmin et al., 2019; Wen et al., 2016; Lasby et al., 2023), could also be realized in the continuous sparsification setting, for instance, by learning neuron-, group, or even layer-wise masks. Yet, this would not enjoy the same theoretical benefits as we derive here by showing that the unstructured continuous $m \odot w$ parameterization induces a mirror flow, whereas for example the neuron-wise mask does not (see Section D).

**Iterative pruning.** Iterative pruning is often motivated by the Lottery Ticket Hypothesis (LTH) (Frankle & Carbin, 2018), which conjectures the existence of sparse subnetworks of larger dense source networks that can achieve the same accuracy as the dense network when trained (Frankle et al., 2021; Liu et al., 2024; Malach et al., 2020; Orseau et al., 2020; Pensia et al., 2020; Burkholz et al., 2022; Fischer & Burkholz, 2021; Burkholz, 2022a;b; da Cunha et al., 2022; Ferbach et al., 2022). In addition to the sparse structure, iterative pruning tries to identify a trainable parameter initialization, indirectly also implementing an approximate $L_0$-regularization. In repeated prune-train iterations, trained weights are thresholded according to an importance score like magnitude. Afterward, the remaining parameters are free to adapt to data in a new training run and not be regularized by a sparsity penalty (like $L_1$). Our proposal PILoT can be combined with such iterative schemes. The experiments show that it boosts performance of state-of-the-art schemes like Weight Rewinding (WR) (Frankle et al., 2019), and Learning Rate Rewinding (LRR) (Maene et al., 2021).

**Continuous sparsification.** Continuous sparsification characterizes a collection of methods that can compete with iterative pruning techniques, while often requiring fewer training epochs (Sreenivasan et al., 2022; Kusupati et al., 2020). In one of the first proposals by (Savarese et al., 2021), the mask

is relaxed to a continuous variable. In general, continuous sparsification can be combined with a probabilistic approach where $m$ is interpreted as a probability (Louizos et al., 2018; Zhou et al., 2021a;b). Other parameterizations of $m$ that are not restricted to the range $[0, 1]$ (e.g. Powerpropagation) can also be utilized to regularize towards higher sparsity (Schwarz et al., 2021). Yet, they are usually combined with projection to map $m$ to a binary mask. The spred algorithm (Ziyin & Wang, 2023) removes any projections and shows that $m \odot w$ with weight decay solve a LASSO objective. To explain its performance gain over LASSO, we extend the mirror flow framework and find an explanation in the training dynamics. Our extension, PILoT, dynamically adjusts the weight decay and induces a transition from an implicit $L_2$ to $L_1$ regularization. This enables it to outperform the state-of-the-art method STR (Kusupati et al., 2020) in the high-sparsity regime. For a survey of other methods see (Kuznedelev et al., 2023).

**Implicit bias.** The implicit bias of (S)GD is a well-studied phenomenon (Chizat & Bach, 2020; Li et al., 2022; Woodworth et al., 2020; Gunasekar et al., 2020; 2017; Chou et al., 2024; Vaškevičius et al., 2019; Li et al., 2023; Zhao et al., 2022; Li et al., 2021) and can in certain cases be described by a mirror flow or mirror descent (in the discrete case with finite learning rate) (Li et al., 2022). Originally, mirror descent was proposed to generalize gradient descent and other first-order methods in convex optimization (Alvarez et al., 2004; Rockafellar & Fenchel, 1970; Boyd & Vandenberghe, 2009; Nemirovski & Yudin, 1983; Beck & Teboulle, 2003). Moreover, it has been used to study the implicit regularization of SGD in diagonal linear networks (Pesme et al., 2021; Even et al., 2023). More recently, it also has been applied to analyze the implicit bias of attention (Sheen et al., 2024). While (Li et al., 2022) has shown that different parameterizations have a corresponding mirror flow, we find that $m \odot w$ with our proposed explicit regularization, PILoT, gives rise to a corresponding time-dependent mirror flow. Its time dependence gives us means to control the implicit bias, while still achieving convergence. Time-dependent mirror descent has so far only been studied in the discrete case as a general possibility (Radhakrishnan et al., 2021). The time dependence also naturally arises in SDE modelling, yet, without control of the implicit bias (Pesme et al., 2021; Even et al., 2023). Here, we not only highlight a practical use case for time-dependent Bregman potentials, we also derive a way to control and exploit it.

**Optimization and convergence proofs.** Loss landscapes and the convergence of first-order methods is a large field of study (Karimi et al., 2016; Fehrman et al., 2019) in its own right. We draw on literature that shows convergence by using the Polyak-Łojasiewicz inequality (Wojtowytsch, 2021; Dereich & Kassing, 2024), which is a more realistic assumption in the deep learning context than, for example convexity, because it can hold locally true for non-convex loss functions that are common in machine learning.

## 2 CONTROLLING THE IMPLICIT BIAS WITH EXPLICIT REGULARIZATION

**Structure of theoretical exposition.** Our first goal is to advance the mirror flow framework from (Li et al., 2022) to incorporate time-dependent regularization. This is a key innovation that forms the foundation of our proposed PILoT algorithm (Algorithm 1). The dynamical description also covers constant regularization, as implemented in the spred algorithm. We begin by integrating time-dependent regularization in the mirror flow framework in the case of the parameterization $m \odot w$ combined with time-varying weight decay. This corresponds to a time-dependent Bregman potential, enabling a more dynamic and powerful form of implicit regularization. The implicit regularization becomes controllable and moves from an $L_2$ to an $L_1$ regularization. Building on this, Theorem 2.1 rigorously characterizes this process within this extended framework, offering new insights into the sparsification process. Then Theorem 2.2 establishes convergence to a solution of the original optimization problem. Sparsity is still attained according to Theorem 2.1, which can also be observed in the gradient flow in Eq. (6) of PILoT. For diagonal linear networks, which is an analytically tractable setting, we also prove optimality, as stated by Theorem 2.3. This highlights a mechanism how our method PILoT improves over spred, since spred cannot reach optimality.

**Optimization problem.** Consider the following time-dependent optimization problem for a loss function $f : \mathbb{R}^n \to \mathbb{R}$:

$$\min_{m,w\in\mathbb{R}^n} f(m \odot w) + \alpha_t \left( ||m||^2_{L_2} + ||w||^2_{L_2} \right). \tag{2}$$

where $\alpha_t \geq 0$ can change during training. In contrast, (Ziyin & Wang, 2023) set $\alpha_t = \alpha$ constant and show that Eq. (2) is equivalent to the LASSO objective. Why does spred tend to outperform LASSO then?

**Seeking answers in the training dynamics.** The gradient flow associated with minimizing the continuously differentiable loss function $f$ is: $dx_t = -\nabla f(x_t)dt$, $x_0 = x_{\text{init}}$. Using this gradient flow framework, (Li et al., 2022) show that a reparameterization or overparameterization of the parameters $x$ leads to a mirror flow. A mirror flow informally minimizes a potential in the background, for example, the $L_1$ or $L_2$-norm. In contrast, explicit regularization forces a direct trade-off. The no need for a trade-off becomes clear in the convergence and optimality theorem.

**Mirror flow.** Concretely, to define a mirror flow, let $R : \mathbb{R}^n \to \mathbb{R}$ be a differentiable function. It is described by
$$d\nabla R(x_t) = -\nabla f(x_t)dt, \qquad x_0 = x_{\text{init}}.$$
(Li et al., 2022) provide sufficient conditions for a paramerization $g: M \to \mathbb{R}^n$ to induce a mirror flow, where $M$ is a smooth sub-manifold in $\mathbb{R}^D$ for $D \geq n$. The parameterization $m \odot w$ falls into this category. The corresponding potential $R$ is either close to the $L_1$ or $L_2$ norm depending on the initialization of $m_0$ and $w_0$. If we could steer it towards $L_1$, we could therefore induce an implicit regularization towards sparsity, yet, not without issues.

**Two caveats and their solution.** a) The potential $R$ attains its global minimum at the initial $x_0$ (and not 0). This also holds for other reparameterizations (Li et al., 2022). In consequence, we would not promote actual sparsity for $x_0 \neq 0$. b) To induce $L_1$ regularization and enter the rich regime, the initialization of both $m_0$ and $w_0$ would need to be exponentially small (Woodworth et al., 2020).

The explicit dynamic regularization of PILoT in Eq. (2) solves both of these problems, as we show next. The corresponding mirror function $R$ is stated in Theorem A.1 and the corresponding convergence and optimality theorems in Theorem A.2 and A.4.

**Dynamic regularization.** Next we present the main result, the dynamical description with time-dependent regularization. The exact dynamics are described by the time-dependent mirror flow and is derived in Theorem 2.1.

**Theorem 2.1** *Let $|w_{0,i}| < m_{0,i}$ for all $i \in [n]$, the time-dependent Bregman potential is given by*

$$R_{a_t}(x) = \frac{1}{2} \sum_{i=1}^{n} x_i \text{arcsinh}\left(\frac{x_i}{a_{t,i}}\right) - \sqrt{x_i^2 + a_{t,i}^2} - x_i \log\left(\frac{u_{0,i}}{v_{0,i}}\right), \tag{3}$$

*with $a_{t,i} = 2u_{0,i}v_{0,i}\exp\left(-2\int_0^t \alpha_s ds\right)$ and $u_{0,i} = \frac{m_{0,i}+w_{0,i}}{\sqrt{2}}$ and $v_{0,i} = \frac{m_{0,i}-w_{0,i}}{\sqrt{2}}$. The gradient flow of $x_t = m_t \odot w_t$ induced by Eq. (2) then satisfies*

$$d\nabla R_{a_t}(x_t) = -\nabla f(x_t)dt, \qquad x_0 = m_0 \odot w_0.$$

**Proof.** The proof is given in the appendix. The main steps are: a) Deriving the evolution of the gradient flow (Lemma B.1). b) Showing that it satisfies the time-dependent mirror flow (Lemma B.2). Note that step a) also derives Eq. (6).

Observe that the potential in Eq. (3) now depends on $a_t$. This changes the global minimum to

$$\nabla R_{a_t}(x) = 0 \Leftrightarrow x = \exp\left(-2\int_0^t \alpha_s ds\right) \odot m_0 \odot w_0.$$

Thus, we gain control over the positional implicit bias, solving our problem with the nonzero global minimum. Next, we characterize the asymptotic behavior, which we control in practice with $\alpha_t$, which determines also $a_t$. The asymptotics follows from Theorem 2 in (Woodworth et al., 2020). For $a \to 0$ and $\left|\frac{x}{a}\right| \to \infty$, we receive

$$R_a(x) \sim \log\left(\frac{1}{a}\right) ||x||_{L_1}.$$

Interestingly, the term $x_i \log\left(\frac{u_{0,i}}{v_{0,i}}\right)$ does not play a role in the asymptotics. The reason is that $\log(\frac{1}{a})$ in front of the other term dominates. Figure 1 illustrates the asymptotics for the one dimensional case. We observe that our two previously identified mirror flow caveats can be resolved: a)

Increasing $a$ moves the minimum to the origin, leading to an $L_1$ regularization. This implies initializing at zero with an exponentially small scaling (i.e. $a \to 0$) is not necessary. b) Moreover, the regularization $\alpha_t$ has an exponential and time-dependent effect on $a_t$. It thus enables steering the dynamics towards an $L_1$ regularization at the desired speed. In conclusion, our extension and novel analysis have revealed how explicit regularization solves the problems of the standard mirror flow framework.

**Remark 2.1** *At the end of LASSO training, the regularization can be turned off to enable the search for a better solution. However, this risks losing the benefits of the regularization, for example, if the basin of attraction contains non-sparse critical points. In contrast, for the $m \odot w$ reparameterization, the time-dependent Bregman potential steers (but does not force) the bias towards sparsity.*

**Convergence and optimality.** It remains to be shown that convergence and optimality results transfer from the mirror flow framework to the time-dependent mirror flow framework. For quasi-convex loss functions, we prove convergence to a critical point. For convex or quasi-convex functions that satisfy the PL-inequality, we derive convergence to a minimizer.

**Theorem 2.2** *Assume $f$ is quasi-convex, $\nabla f$ is locally Lipschitz and $\mathrm{argmin}\{f(x)|x \in \mathbb{R}^n\}$ is non-empty. Assume $\alpha_t \geq 0$ for all $t \geq 0$ and that $\int_0^t \alpha_s ds < \infty$ for $t \in [0,\infty) \cup \{\infty\}$. Then as $t \to \infty$, $x_t$ converges to some critical point $x^*$. Furthermore, if $f$ is either convex or both quasi-convex and satisfies the PL-inequality in Eq. (9). Then $x_t$ convergences to an interpolator $x^*$ that is a minimizer of $f$. Furthermore, in the PL-inequality case, the loss converges linearly such that there is a constant $C > 0$ such that*

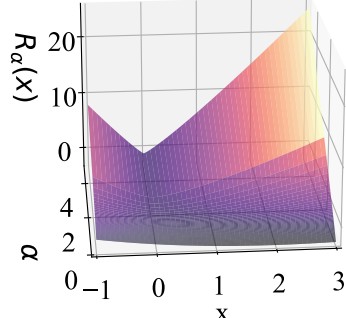

Figure 1: Evolution of the time-dependent Bregman potential. $\alpha = \int_0^t \alpha_s ds$ is the exponent of $a_t$.

$$f(x_t) - f(x^*) \leq B exp\left(-\lambda A_\infty t\right), \qquad (4)$$

*where $B = (f(x_0) - f(x^*)) \, exp\left(C\|x^*\|_{L_2} \int_0^\infty \alpha_s ds\right)$ with $C$ depending on the smoothness of the loss function and $A_\infty = \min_i \left(m_{0,i}^2 - w_{0,i}^2\right) \exp\left(-2 \int_0^\infty \alpha_s ds\right)$.*

**Proof.** The main steps of the proof are to show that a) the iterates are bounded and converge to a critical point (Lemma B.3); b) the loss converges (Theorem B.1). A noteworthy tool is a time-dependent Bregman divergence, which we use to bound the iterates. Furthermore, we utilize that $\alpha_t \geq 0$ converges to zero, resulting eventually in a non-increasing evolution of the loss.

**Potential drawbacks.** Theorem 2.2 guarantees convergence in the case of implicit regularization. Explicit $L_1$ regularization or spred, on the other hand, cannot achieve the same result due to constant regularization, which we will also highlight in experiments. Regardless, note that the constant $B$ in Eq. (4) could be large. Furthermore, to reach the implicit $L_1$-regularization, $a_t = \left(m_0^2 - w_0^2\right) \exp\left(-2 \int_0^t \alpha_s ds\right)$ needs to be exponentially small similarly as in Theorem 2 in (Woodworth et al., 2020). These two potential drawbacks also reveal where the method will work, namely, in overparameterized settings where the solution $x^*$ should have less active parameters. Then $B$ is potentially relatively small.

**Remark 2.2** *If $\nabla f$ is one-sided inversely Lipschitz, a speed-up is possible. The quantity that needs to be bounded for convergence is $-\nabla f(x_t)^\top x_t$. In this case, we get*

$$-\nabla f(x_t)^\top x_t \leq -\nabla f(x_t)^\top x^* - \|x_t - x^*\|_{L_2}^2 \leq C\|x^*\|_{L_2} - \|x_t - x^*\|_{L_2}^2,$$

*where $C$ is the bound on the smoothness of the loss function $f$. This implies that when the interpolator $x^* \approx 0$ is small, the right-hand side is negative, leading to a speed-up. This condition is also known as coercive.*

**Optimality.** The main assumption in Theorem 2.2 is $\alpha_t \to 0$, which ensures convergence to a minimizer of the original problem, while retaining sparsity depending on $a_t \to a_\infty$. In the case of diagonal linear networks, we can even prove optimality with respect to the final Bregman potential $R_{a_\infty}$.

**Theorem 2.3** *In case of under-determined regression consider the loss function $f(x) = \tilde{f}(Zx - Y)$. Assume $f$ satisfies the conditions with at least one of the convergence criteria of Theorem 2.2. Then $x_t$ converges to $x^*$ such that*

$$x^* = argmin_{Zx=Y} R_{a_\infty}(x). \tag{5}$$

**Proof.** We show that the KKT conditions of Problem (5) are satisfied (Theorem B.2).

**Take away.** We have shown that our extended mirror flow framework can attain convergence and optimality in an analytically tractable scenario. Furthermore, from a practical side, it also gives us new tools to derive more promising continuous sparsification techniques with implicit regularization. In the following section, we propose a way to dynamically control the transition from implicit $L_2$ regularization to $L_1$ during training with the help of the derived time-dependent Bregman potential.

## 3 THE ALGORITHM: PILoT

Like spred, our new algorithm PILoT (Algorithm 1) utilizes the parameterization $m \odot w$, but proposes a novel initialization and dynamic regularization schedule to control the transition from implicit $L_2$ to $L_1$ regularization. To attain the desired results in the original parameterization $x$, we first derive its gradient flow.

**Gradient flow.** Essentially, the gradient flow follows from the analysis in Section 2. According to Theorem 2.2, we guarantee convergence by ensuring $\alpha_t \to 0$. Concretely, the gradient flow for $x_t = m_t \odot w_t$ induced by Eq. (2) is given by:

$$dx_t = -\sqrt{x_t^2 + a_t^2} \odot \left( \nabla f(x_t) + 2\alpha_t \frac{x_t}{\sqrt{x_t^2 + a_t^2}} \right) dt, \qquad x_0 = x_{init}, \tag{6}$$

where $a_t = \left( m_0^2 - w_0^2 \right) \exp\left( -2 \int_0^t \alpha_s ds \right)$. $m_0$ and $w_0$ have to be initialized such that $m_0 \odot w_0 = x_{init}$. Note that all operations are point-wise. The derivation is based on the time-dependent mirror flow in Section 2.

**Remark 3.1** *The gradient flow in Eq. (6) allows us to make a direct comparison to the continuous sparsification method STR (Kusupati et al., 2020). Instead of the soft thresholding operator, we have $\sqrt{x_t^2 + a_t^2}$. The main difference is that STR does not change the magnitude of the gradient update outside of the (learnable) threshold, while both PILoT and spred actively change the magnitude depending on the magnitude of the weight. This active sparsification explains why spred and also PILoT can perform better in the high-sparsity regime.*

**Spred.** The gradient flow in Eq. (6) explains why spred performs better than LASSO and highlights where spred can further be improved. Note that the balanced initialization of spred is defined such that $m_0^2 - w_0^2 = 0$ and the regularization is constant $\alpha_t = \alpha$. Plugging this into Eq. (6) gives

$$dx_t = -\sqrt{x_t^2} \odot \left( \nabla f(x_t) + 2\alpha \text{sign}(x_t) \right) dt, \qquad x_0 = x_{init}. \tag{7}$$

Compare this with the gradient flow of LASSO with regularization strength $2\alpha$:

$$dx_t = -\left( \nabla f(x_t) + 2\alpha \text{sign}(x_t) \right) dt, \qquad x_0 = x_{init}.$$

We observe that the main difference to the gradient flow of LASSO is the factor $\sqrt{x_t^2}$. This implies the considerable drawback that spred gradient flows cannot sign flip. Therefore, it cannot reach the optimal solution or specific minimizers potentially. Another way to see this is studying the evolution $x_t = x_0 \exp\left( -4\text{sign}(x_0) \int_0^t \nabla f(x_s) ds - 2\alpha t \right)$ satisfying Eq. (7). In practice, the absence of sign flips might be remedied by using a large learning rate and noise. The evolution also explains why it can perform better than LASSO, as it decays redundant parameters exponentially faster instead of linearly. In other words, the gradient update is proportional to the magnitude of the parameter. Therefore, the evolution of spred (Eq. (7)) can converge faster and come closer to zero than LASSO.

**PILoT.** Our main goal is to remedy the caveats of the spred by inducing the more general gradient flow in Eq. (6). The first improvement is to enable sign flips by changing the initialization to $m_0^2 -$

$w_0^2 = \beta > 0$, where $\beta$ denotes the scaling constant. After discretizing Eq. (6), the effective learning rate at initialization $x_0 = 0$ is $\eta|\beta|$, where $\eta > 0$ is the learning rate. Therefore, we use $\beta = 1$ in the experiments so that the learning rate $\eta$ is not altered.

Our second and main improvement is induced by the time dependence of $\alpha_t$. $\alpha_t$ controls the strength of both the implicit and explicit regularization via $a_t$. If $a_t >> x_t$, then the regularization term in Eq. (6) resembles an $L_2$ instead of an $L_1$ norm. Therefore decreasing $a_t$ moves the implicit regularization from $L_2$ to $L_1$. Accordingly, we sparsify only mildly in early training epochs in contrast to spred. We have shown this formally in Section 2. Furthermore, convergence is covered by Theorem 2.2 when $\alpha_t \to 0$. Even when PILoT attains a similar sparsity as spred at the end of the training dynamics, it can usually still achieve a higher accuracy due to its improved training dynamics.

**Details on PILoT.** The described design choices define Algorithm 1. Our update of the regularization strength $\alpha_k$ depends on three quantities: a) the sparsity threshold $K$ for the weights (a hyperparameter), b) the training accuracy, and c) $\delta \geq 1$, the multiplicative factor to gradually increase or decrease the regularization strength. The regularization strength (and thus sparsity) grows if the sparsity threshold has not been reached yet and the training accuracy has increased in the previous gradient update step. As the strength is adaptive, the algorithm is less sensitive to the initial strength $\alpha_0$. Note that the setting $\delta = 1$ and $\beta = 0$ corresponds to spred. Therefore, PILoT is a strict generalization of spred. In contrast to spred, however, after half of the training epochs, we decay the regularization strength regardless of whether the sparsity threshold $K$ is reached. This guarantees convergence of the corresponding gradient flow, in accordance with Theorem 2.2.

---

**Algorithm 1** PILoT

---

**Require:** epochs $T$, schedule $\alpha_{init}$, initialization $x_{init}$, scaling constant $\beta$
    Initialize $m_0, w_0$ such that $m_0 \odot w_0 = x_{init}$, $m_0^2 - w_0^2 = \beta$, $\delta \geq 1$ and, $K$
    $\alpha_0 \leftarrow \alpha_{init}$
    $Current\_training\_acc \leftarrow 0$
    Set $\tilde{f}(m, w, \alpha_0) := f(m \odot w) + \alpha_0 \left( ||m||_{L_2}^2 + ||w||_{L_2}^2 \right)$
    **for** $k$ in $1 \ldots T$ **do**
        $(m_k, w_k) = \text{OptimizerStep}\left( \tilde{f}(m_{k-1}, w_{k-1}, \alpha_{k-1}) \right)$
        **if** $Training\_acc \geq Current\_training\_acc$ and $||m_k \odot w_k||_{L_1} \geq K$ and $k \leq \frac{T}{2}$ **then**
            $\alpha_k \leftarrow \alpha_{k-1}\delta$
        **else**
            $\alpha_k \leftarrow \alpha_{k-1}/\delta$
        **end if**
        $Current\_training\_acc \leftarrow Training\_acc$
    **end for**
    **return** Model $f(x_T)$ with $x_T = m_T \odot w_T$

---

## 4 EXPERIMENTS

We demonstrate the effectiveness of PILoT in extensive experiments covering three different scenarios. Firstly, we confirm our theoretical results on the gradient flow in Theorem 2.3. Secondly, we compare PILoT with other state-of-the-art continuous sparsification methods such as STR (Kusupati et al., 2020) and spred (Ziyin & Wang, 2023) in a one-shot setting. In this context, we also isolate the individual contribution of our initialization. Finally, we combine PILoT with iterative pruning methods such as WR (Frankle & Carbin, 2019) and LRR (Maene et al., 2021).

**Memory requirements.** As most other continuous sparsification approaches, note that PILoT doubles the number of parameters during training. Yet, according to Ziyin & Wang (2023), the training time of a ResNet50 with $m \odot w$ parameterization on ImageNet increases roughly by 5% only and the memory cost is negligible if the batch size is larger than 50. At inference, we would return to the original representation $x$ and therefore benefit from the improved sparsification.

**Diagonal Linear Network.** We have proven optimality for the analytically tractable setting of diagonal linear networks. Now we illustrate the benefit of our initialization and dynamic explicit regularization. Furthermore, we highlight the impact of a good dynamic schedule of the regularization strength $\alpha_t$. We use $d = 40$ amount of data points with feature dimension $n = 100$ and sample $z_j \sim N(0, \mathbb{I}_n)$ for $j \in [d]$. The ground truth $x^*$ is set such that $||x^*||_{L_0} = 5$. Furthermore, the network parameters are initialized with $x_0 \sim N(0, \mathbb{I}_n \frac{1}{\sqrt{n}})$. The step-size is $\eta = 10^{-4}$ and the trajectories are averaged over 5 initializations. 0.95 confidence regions are indicated by shades. The mean squared error is used as loss function. We report the distance to the ground truth $||x_t - x^*||_{L_2}$ over training time in Figure 2. Two different initializations, i.e., the one of spred and of PILoT, and different regularization schedules are considered. The schedules are described in Appendix C.1. Results for the best ones are shown in the figure and confirm our theoretical insights. The inability to sign flip prevents spred from reaching the ground truth for all considered schedules. Furthermore, with a dynamic regularization, our PILoT initialization outperforms both spred and LASSO, reaching the ground truth. The best-performing schedule for the PILoT initialization is a geometrically decaying schedule, as also implemented in Algorithm 1. In contrast, for the other two methods, a constant regularization works best. This experimentally confirms Theorem 2.3 and Remark 2.2. Note that LASSO with gradient descent performs as expected. Decaying schedules perform worse than constant ones supporting Remark 2.1.

**One-shot sparsification.** Firstly, we compare our method PILoT with STR, spred, and LASSO on CIFAR10 and CIFAR100 training a ResNet-20 or ResNet-18, respectively. Furthermore, in the case of CIFAR10, we also implement the novel initialization ($m_0^2 - w_0^2 = 1$) without dynamic regularization to isolate its benefits. We consider two learning rates $\{0.1, 0.2\}$ and the weight decay range $\{1e-5, \dots 1e-2\}$ for CIFAR10 and range $\{1e-4, \dots 1e-3\}$ for CIFAR100 and always show the best result. The same range for the regularization strength is explored for LASSO. The other hyperparameters are reported in the appendix. Secondly, we train ResNet-50 on ImageNet with the setup of STR (Kusupati et al., 2020) and compare directly with their results. Furthermore, we implement both PILoT and spred in this setting.

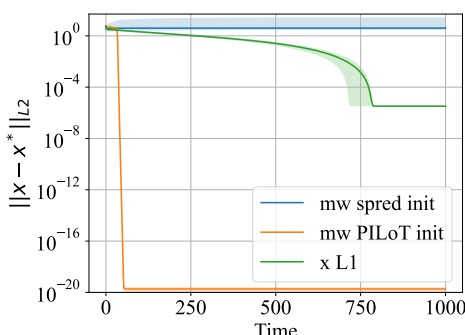

Figure 2: A simulation of gradient flow on a diagonal linear network is given for the different regularizations.

Figure 3 presents results for CIFAR10 and CIFAR100. PILoT outperforms all other methods and is particularly effective in the high-sparsity regime. Our PILoT initialization leads to improvements over spred and STR for medium levels of sparsity. This supports our theoretical insight into the role of initializations and how they influence the implicit bias. In addition, STR is outperformed by spred in the high-sparsity regime, confirming the findings of (Ziyin & Wang, 2023).

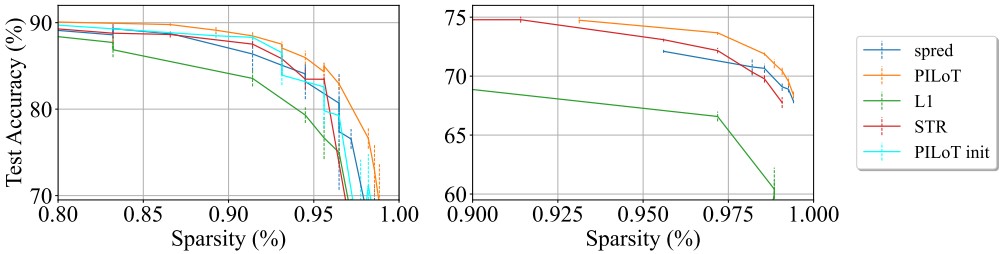

Figure 3: One-shot sparsification. Acc. versus sparsity for CIFAR10 (left) and CIFAR100 (right).

In Table 1, we compare PILoT to both STR and spred on ImageNet (Deng et al., 2009). See Appendix C.2 Table 3 for details on experimental configurations. Our method competes with or out-

---

[1]Starting from a pretrained model with 77% validation accuracy

Table 1: ResNet-50 on ImageNet sparsity (%) versus accuracy (%) results.

| Method | Top-1 Acc | Sparsity |
|---|---|---|
| ResNet-50 | 77.01 | 0 |
| STR | **76.19** | 79.55 |
| STR | 76.12 | **81.27** |
| spred[1] | 75.5 | 80.00 |
| spred | 72.64 | 79.03 |
| PILoT | 75.62 | 80.00 |
| STR | **74.73** | 87.7 |
| STR | 74.01 | 90.55 |
| spred | 71.84 | 89.26 |
| PILoT | **74.73** | 88.00 |
| PILoT | 74.04 | **91.00** |

| Method | Top-1 Acc | Sparsity |
|---|---|---|
| ResNet-50 | 77.01 | 0 |
| STR | 70.4 | 95.03 |
| spred | 69.47 | 94.50 |
| PILoT | **72.67** | 94.00 |
| PILoT | **71.30** | 95.00 |
| PILoT | **71.05** | **95.60** |
| PILoT | **70.49** | **96.00** |
| STR | 67.22 | 96.53 |
| spred | 66.12 | **97.19** |
| PILoT | **68.49** | **97.19** |
| STR | 61.46 | 98.05 |
| spred | 62.71 | 98.20 |
| PILoT | **66.49** | 97.75 |
| PILoT | **64.06** | **98.20** |

Figure 4: Learning Rate Rewinding (LRR) and Weight Rewinding (WR) with PILoT on ImageNet ResNet-18. The left plot is the complete plot and the right plot is a zoomed-in version.

performs all baselines at medium and high sparsity levels. In addition, it improves over spred at 80% sparsity, even when spred is initialized with a 77% pretrained ResNet-50. This highlights the effectiveness of PILoT and confirms the insights from the developed theory.

**Iterative Pruning.** To demonstrate the versatility of PILoT, we also combine it with the state-of-the-art iterative pruning methods Learning Rate Rewinding (LRR) (Maene et al., 2021) and Weight Rewinding (WR) (Frankle et al., 2019) on ImageNet using a ResNet-18. For simplicity, we use $\beta = 1$ and no regularization. We see in Figure 4 that the parameterization $m \odot w$ futher boosts the performance of iterative methods. Remarkably, WR becomes competitive with LRR. Additional experiments on CIFAR10 and CIFAR100 with regularization are in Appendix C.3. The regularization further helps to reach higher accuracy at high sparsities.

## 5 DISCUSSION

We have shed light on the inner workings of continuous sparsification. Its basic relaxed formulation utilizes the parameterization $m \odot w$, which induces an implicit bias towards sparsity. In contrast to an explicit $L_1$-regularization, it enjoys all the benefits of an implicit regularization that caters first to the loss and not a sparsity penalty. Exploiting this insight for neural network sparsification, we have proposed PILoT, which relies on a controllable regularization that acts like an implicit regularization in the original neural network parameter space and, remarkably, corresponds to a time-dependent Bregman potential. As we have shown, the time-dependent control enables the associated mirror flow to enter the so-called rich regime, and thus effectively change the implicit regularization from $L_2$ to $L_1$. This property is central to showing convergence of our approach for (quasi)-convex loss functions and optimality for underdetermined linear regression. Experiments on standard vision benchmarks further corroborate the utility of our theoretical insights, as our proposal PILoT achieves significant improvements over state-of-the-art baselines.

ACKNOWLEDGMENTS AND DISCLOSURE OF FUNDING

The authors gratefully acknowledge the Gauss Centre for Supercomputing e.V. for funding this project by providing computing time on the GCS Supercomputer JUWELS at Jülich Supercomputing Centre (JSC). We also gratefully acknowledge funding from the European Research Council (ERC) under the Horizon Europe Framework Programme (HORIZON) for proposal number 101116395 SPARSE-ML.

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

## A   MIRROR FLOW FRAMEWORK

In this section we present some known results from the mirror flow framework for completeness. We derive the Bregman potential associated with $m \odot w$ in Theorem A.1. Next, we will also show the convergence of the loss and provide optimality guarantees in Theorem A.2. In addition, we extend Theorem A.2 with Theorem A.3. Finally, we give the optimality result for diagonal linear networks in Theorem A.4

**Theorem A.1** *Let the initialization of $m$ and $w$ satisfy $m_{0,i} > |w_{0,i}|$ for all $i \in [n]$. Then the corresponding mirror function is:*

$$R(x) := \frac{1}{4}\sum_{i=1}^{n} x_i arcsinh\left(\frac{x_i}{2u_{0,i}v_{0,i}}\right) - \sqrt{x_i^2 + 4u_{0,i}^2 v_{0,i}^2} - x_i log\left(\frac{u_{0,i}}{v_{0,i}}\right) \tag{8}$$

*where $u_{0,i} = \frac{m_{0,i}+w_{0,i}}{\sqrt{2}}$ and $v_{0,i} = \frac{m_{0,i}-w_{0,i}}{\sqrt{2}}$. Furthermore, $R$ is a Bregman function.*

Proof. The result follows directly from applying Theorem 4.16 in (Li et al., 2022).

Theorem A.1 implies the following: a) The global minima of $R$ is at the initialization $x_0 = m_0 \odot w_0$. b) The Lipschitz coeficient of $R$ depends on the initalization. The Lipschitz coeficient $L_R$ of (8) is $L_R = \frac{1}{\min_i 2u_{0,i}v_{0,i}}$, determining the smoothness of the potential. Following these two observations we make the following remark about Theorem A.1.

**Remark A.1** *Note that when the initialization is zero, i.e., $w_0 = 0, m_0 = \sqrt{a}$ with $a \geq 0$ then (8) is the hyperbolic entropy. The hyperbolic entropy is*

$$\sum_{i=1}^n x_i arcsinh\left(\frac{x_i}{a}\right) - \sqrt{x_i^2 + a^2}$$

*Theorem 2 of (Woodworth et al., 2020) characterizes the behavior in the limit for this case. For the hyperbolic entropy in case $a \to 0$ and $|\frac{x}{a}| \to \infty$,*

$$R(x) \sim log\left(\frac{1}{a}\right) ||x||_{L_1}.$$

*This means an $L_1$ bias is induced when $a$ is small. Nevertheless, we need an exponentially small $a$ compared to $x$ to get there as shown in (Woodworth et al., 2020), which can lead to numerical problems. Furthermore, $m_0 = w_0 = 0$ is a saddle point which can slow down training (exponentially) (Du et al., 2017). Additionally, the asymptotic result only holds for initializing at zero. Note $L_R = a$ in this case.*

Remark A.1 shows the potential of using the implicit bias to induce sparsity. To actualize this, we need to solve the two challenges posed in the remark. Both are remedied in Section 2.

In addition to this promising formulation of implicitly minimizing an $L_1$ norm with the use of the mirror framework, we can get convergence results. These results make it clear why implicit regularization is preferable over explicit regularization. The convergence result from (Li et al., 2022) is stated for our setting. Furthermore, the theorem is extended for a specific class of Bregman functions.

**Theorem A.2** *(Theorem 4.14 (Li et al., 2022)) Assume that $f$ is quasi-convex, $\nabla f$ is locally Lipschitz and $argmin\{f(x)|x \in \mathbb{R}^n\}$ is non-empty. Then as $t \to \infty$, $x_t$ converges to some critical point $x^*$. Moreover, if $f$ is convex $x_t$ converges to a minimizer of $f$.*

In Theorem A.2 it is shown that with implicit regularization an optimal solution to the original optimization problem can be reached. In contrast, explicit regularization makes this not possible, by definition. Because the optimization problem has fundamentally changed. Showing the benefit of implicit over explicit.

For the extension, the convexity constraint is replaced by the Polyak-Łojasiewicz (PL) inequality in the theorem. The PL-inequality is a more realistic constraint in a machine learning context as loss functions are not locally convex but can satisfy the PL inequality locally (Wojtowytsch, 2021; Dereich & Kassing, 2024). The PL-inequality for a continuously differentiable function $f$ is

$$||\nabla f(x)||_{L_2}^2 \geq \lambda\left(f(x) - f(x^*)\right) \qquad \forall x \in \mathbb{R}^n \tag{9}$$

for some $\lambda > 0$ and global minima $x^*$ of $f$. This allows us to state the modified theorem.

**Theorem A.3** *Consider the same setting as Theorem A.2. Assume $R$ satisfies for all $x \in \mathbb{R}^n$,*

$$z^T \left(\nabla^2 R(x)\right)^{-1} z \geq \sigma ||z||_{L_2}^2 \qquad \forall z \in \mathbb{R}^n. \tag{10}$$

*Furthermore, assume $f$ satisfies the PL-inequality (9). Then $x_t$ converges to a minimizer of $f$. Furthermore, the loss converges linearly with rate $\sigma\lambda$.*

Proof.      The   evolution   of   $f(x_t) - f(x^*)$   is   described   by   $df(x_t) = -\nabla f(x_t)^\top \left(\nabla^2 R(x_t)\right)^{-1} \nabla f(x_t)dt$. From (10) and (9) the evolution is bounded by

$$df(x_t) \leq -\sigma||\nabla f(x_t)||_{L_2}^2 dt \leq -\sigma\lambda\left(f(x_t) - f(x^*)\right)dt.$$

Applying Gronwall's Lemma concludes the proof. □

Note that Theorem A.3 holds in the same (general) setting as Theorem A.2. Also, note that the PL-inequality together with quasi-convexity does not imply convexity. Theorem A.3 holds for our setting. In this case, it follows from a direct computation that

$$\left(\nabla^2 R(x)\right)^{-1} = \text{diag}\left(\sqrt{x^2 + 4u_{0,1}^2 v_{0,1}^2}, \ldots, \sqrt{x^2 + 4u_{0,n}^2 v_{0,n}^2}\right).$$

This implies that $\sigma = 2\min_i u_{0,i} v_{0,i}$ in Theorem A.3, which again highlights the importance of the initialization.

Finally, in the case of under-determined linear regression, we can derive optimality conditions in the form of KKT conditions of $R$. Consider a data set $(z_j, y_j)_{i=1}^d$ with $z_j \in \mathbb{R}^n$ and $y_j \in \mathbb{R}$. Let $Z = (z_1, \ldots z_d)$ and $Y = (y_1, \ldots y_d)$. For the regression to be called underdetermined $n > d$.

**Theorem A.4** *(Theorem 4.17 (Li et al., 2022)) In case of under-determined regression consider the loss function $f(x) = \tilde{f}(Zx - Y)$. Assume $f$ satisfies the conditions of Theorem A.2. Then $x_t$ converges to $x^*$ such that*

$$x^* = argmin_{Zx=Y} R(x)$$

Note that Theorem 4.17 of (Li et al., 2022) only uses quasi-convexity of the loss. Theorem A.4 guarantees that the optimization problem is solved while implicitly minimizing the potential $R$. Thus choosing the sparsest model out of the models that predict the data perfectly. This highlights another benefit of implicit regularization over explicit regularization.

In this section, we have shown the viability of using the implicit bias framework to induce an implicit regularization. Furthermore, we have given two known benefits of using the implicit bias framework over explicit regularization. The benefits are convergence to the optimal solution of the original problem and optimality in the case of underdetermined regression. To add to this, we have extended the convergence theorem using the PL-inequality in 9. Moreover, we again highlight the importance of the initialization of $m_0$ and $w_0$ with the influence on the smoothness of the Bregman potential and convergence of the loss. The initialization insight as in the main text is used to improve upon spred (Ziyin, 2023) as their initialization has scaling $2u_0 v_0 = 0$, it follows already from the mirror flow framework that $2u_0 v_0 = 1$ is a better initialization. Furthermore, we also do not initialize at zero though, and scaling needs to be exponentially small to get a good approximation of the $L_1$ norm potentially making it hard to escape the saddle point. Therefore, the explicit regularization analyzed in Section 2 is necessary to exploit the implicit bias framework.

## B  PROOF MAIN RESULT

We show the main result here. The proof consists of four parts

- $R_{a_t}$ satisfies a mirror flow (Lemmas B.1 and B.2)
- Boundedness of the iterates and convergence to a critical point (Lemma B.3)
- Convergence of the loss (Theorem B.1)
- Optimality in case of underdetermined linear regression (Theorem B.2)

Consider the following gradient flow

$$\begin{cases} dm_t = -\nabla f(m_t \odot w_t) \odot w_t - 2\alpha_t m_t dt \\ dw_t = -\nabla f(m_t \odot w_t) \odot m_t - 2\alpha_t w_t dt \end{cases} \tag{11}$$

For the flow in (11) to be well-posed $\nabla f$ needs to be locally Lipschitz continuous. This is a sufficient condition given that $\alpha_t$ is "nice", which will be made more rigorous later. The evolution of $x_t = m_t \odot w_t$ is derived in Lemma B.1.

**Lemma B.1** *The evolution of $x_t = m_t \odot w_t$ with 11 is described by*

$$x_t = u_0^2 \odot exp\left(-2\int_0^t \nabla f(x_s)\, ds - 4\int_0^t \alpha_s ds\right) - v_0^2 \odot exp\left(2\int_0^t \nabla f(x_s)\, ds - 4\int_0^t \alpha_s ds\right),$$

where $u_0 = \frac{m_0+w_0}{\sqrt{2}}$ and $v_0 = \frac{m_0-w_0}{\sqrt{2}}$.

Proof. This follows from deriving the flow of $m_t$ and $w_t$ and then combining the two. The evolution of both are given by

$$
\begin{cases}
m_t = \left(m_0 \odot \cosh\left(-\int_0^t \nabla f(x_s)\,ds\right) + w_0 \odot \sinh\left(-\int_0^t \nabla f(x_s)\,ds\right)\right) \exp\left(-2\int_0^t \alpha_s ds\right) \\
w_t = \left(w_0 \odot \cosh\left(-\int_0^t \nabla f(x_s)\,ds\right) + m_0 \odot \sinh\left(-\int_0^t \nabla f(x_s)\,ds\right)\right) \exp\left(-2\int_0^t \alpha_s ds\right).
\end{cases}
$$

For ease of notation set $L_t = \int_0^t \nabla f(x_s)\,ds$ and $A_t = \int_0^t \alpha_s ds$. Combining gives us

$$
\begin{aligned}
x_t &= m_t \odot w_t \\
&= \left(m_0^2 + w_0^2\right) \odot \cosh\left(-L_t\right) \odot \sinh\left(-L_t\right) \exp\left(-4A_t\right) \\
&\quad + w_0 \odot m_0 \odot \left(\cosh\left(-L_t\right)^2 + \cosh\left(-L_t\right)^2\right) \exp\left(-4A_t\right) \\
&= \left(\frac{\left(m_0^2 + w_0^2\right)}{2} \odot \sinh\left(-2L_t\right)\right) \exp\left(-4A_t\right) \\
&\quad + w_0 \odot m_0 \odot \left(\cosh\left(-2L_t\right)\right) \exp\left(-4A_t\right) \\
&= u_0^2 \odot \exp\left(-2L_t - 4A_t\right) - v_0^2 \odot \exp\left(2L_t - 4A_t\right)
\end{aligned}
$$

where the second equality follows from hyperbolic identities. $\square$

It follows from Lemma B.1 and the local Lipschitz condition on $\nabla f$ and $\int_0^t \alpha_s ds < \infty$ for all $t \geq 0$, that the flow is well-posed. We now define the (corrected)-hyperbolic entropy function. The corrected hyperbolic entropy is given by

$$
R_a(x) = \frac{1}{2}\sum_{i=1}^{n} x_i \operatorname{arcsinh}\left(\frac{x_i}{a}\right) - \sqrt{x_i^2 + a^2} - x_i \log \frac{u_{0i}}{v_{0,i}},
$$

where the last term is the correction. The correction stems from not initializing at zero.

**Lemma B.2** Let $|w_{i0}| \leq m_{0i}$ for all $i \in [n]$, then $R_{a_t}(x_t)$ with $a_t = 2u_0 \odot v_0 \exp\left(-2\int_0^t \alpha_s ds\right)$ satisfies

$$
d\nabla R_{a_t}(x_t) = -\nabla f(x_t)dt \qquad x_0 = m_0 \odot w_0. \tag{12}
$$

Proof. This follows from Lemma B.1,

$$
x_t \exp\left(4\int_0^t \alpha_s ds\right) = u_0^2 \exp\left(-2\int_0^t \nabla f(x_s)ds\right) - v_0^2 \exp\left(2\int_0^t \nabla f(x_s)ds\right) \Leftrightarrow
$$

$$
\frac{1}{2}\left(\operatorname{arcsinh}(\frac{x_t}{a_t}) - \log(\frac{u_0}{v_0})\right) = -\int_0^t \nabla f(x_s)ds.
$$

This equivalence follows from setting $z = \exp\left(-2\int_0^t \nabla f(x_s)ds\right)$ and solving the resulting quadratic equation. Notice that the left hand side in (12) is $\nabla R_{a_t}(x_t)$. $\square$

**Lemma B.3** Let $f$ be a quasi-convex function and $\alpha_t \geq 0$ for all $t \geq 0$. Furthermore, assume the integral $\int_0^t \alpha_s ds < \infty$. Then the iterates are bounded and converge to a critical point.

Consider the time-dependent Bregman divergence

$$
D_{a_t}(x^*, x_t) := R_{a_t}(x^*) - R_{a_t}(x_t) - \nabla_x R_{a_t}^T(x^* - x_t) \geq 0
$$

The divergence is bounded by:

$$
D_{a_t}(x^*, x_t) \leq R_{a_\infty}(x^*) - R_{a_t}(x_t) - \nabla_x R_{a_t}^T(x^* - x_t) =: W_t,
$$

this follows from the fact that the map $a \to R_a$ is decreasing. We make the following two observations:

$$
\frac{d}{da}R_a(x) = -\frac{1}{2}\frac{x^2|a| + a^3}{a^2\sqrt{a^2 + x^2}} \leq 0 \qquad \text{and} \qquad \frac{da_t}{dt} \leq 0 \qquad \forall t \geq 0. \tag{13}
$$

This allows us to bound the evolution

$$
\begin{aligned}
\frac{d}{dt} W_t &= \frac{d}{dt} \left( -R_{a_t}(x_t) - \nabla_x R_{a_t}^T (x^* - x_t) \right) \\
&= -\frac{d}{da} R_{a_t}(x_t) \frac{d}{dt} a_t - \frac{d}{dt} \left( \nabla_x R_{a_t}^T \right) (x^* - x_t) \\
&\leq \nabla_x f(x_t)^T (x^* - x_t) \\
&\leq 0,
\end{aligned}
$$

where the observations in (13) are used in the first inequality.

From Theorem 4.16 in (Li et al., 2022) it follows that for all $a > 0$, $R_a$ is a Bregman potential. Implying that for all $a > 0$ the level set for $\gamma \in \mathbb{R}$,

$$
\{ x \in \mathbb{R}^n : D_a(x^*, x) \leq \gamma \}
$$

is bounded. Combining this with the fact that the evolution is bounded, implies the iterates are bounded.

In the next part, it is shown that $x_t$ converges to a critical point. We first show that the loss becomes eventually non-increasing. There is a $T$ such that for all $t \geq T$ the loss $f$ is non-increasing.,

$$
\begin{aligned}
df(x_t) &= -\left( \nabla f(x_t)^T \mathrm{diag}\left( \sqrt{x_t^2 + a_t^2} \right) \nabla f(x_t) + 2\alpha_t \nabla f(x_t)^T x_t \right) dt \\
&\leq \left( -\nabla f(x_t)^T \mathrm{diag}\left( \sqrt{x_t^2 + a_t^2} \right) \nabla f(x_t) + 2\alpha_t C \right) dt,
\end{aligned}
$$

where it is used that the iterates are bounded and $\nabla f$ is locally Lipschitz. As $t \to \infty$ we have that $\alpha_t \to 0$ and $a_t \to a_\infty > 0$ by assumption. Hence there exists a $T$ such that for all $t \geq T$ we have

$$
df(x_t) \leq 0.
$$

Note if this is not the case then there is a $T > 0$ such that $\nabla f(x_T) = 0$ implying convergence (to a critical point).

Now let $x_\infty$ be an accumulation point of the bounded flow $x_t$. We use this to show convergence to a critical point. We have that for all $x \in \mathbb{R}^n$,

$$
\nabla f(x_\infty)^\top x = \lim_{t \to \infty} \frac{1}{t} \left( \int_T^{T+t} \nabla f(x_s) ds \right)^\top x = \lim_{t \to \infty} \frac{1}{t} \left( R_{a_T}(x_T) - R_{a_{T+t}}(x_{T+t}) \right)^\top x = 0,
$$

where the first equality follows from that the loss is non-increasing and the second one from the time-dependent mirror flow description. Finally, because $x_t$ converges to an accumulation point we also have $\lim_{t \to \infty} R_{a_t}(x_t) = R_{a_\infty}(x_\infty)$ by continuity, giving the last equality.

We use that the accumulation point is a critical point and set $x^* = x_\infty$ in $W_t$ such that $W_t \to 0$. This implies $D_{a_t}(x_\infty, x_t) \to 0$ by the upperbound. It follows from the fact that the iterates are bounded that $R_{a_t}$ is $\mu$-strongly convex on this bounded convex set where the iterates stay. This gives

$$
||x_\infty - x_t||_{L_2} \leq \frac{\mu}{2} D_{a_t}(x_\infty, x_t) \to 0,
$$

showing $x_t$ converges to a critical point. $\square$

Lemma B.3 gives a condition such that the iterates are bounded and converge to a critical point. It remains to be shown that the loss converges. This is done in Theorem B.1.

**Theorem B.1** *Consider the same setting as Lemma B.3, if $f$ is convex or satisfies the PL-inequality we have convergence to an interpolator $x^*$ such that it is a minimizer of $f$. Furthermore, in the PL-inequality case, the loss converges linearly.*

Proof. Assume $f$ is convex, notice first that there is a $T$ such that for all $t \geq T$ the loss is non-increasing. Combining this with a bound on the time-dependent Bregman potential gives us convergence of the loss. The time-dependent Bregman divergence is again defined by

$$
D_{a_t}(x^*, x_t) = R_{a_t}(x^*) - R_{a_t}(x_t) - \nabla_x R_{a_t}^\top (x^* - x_t) \geq 0.
$$

The divergence is bounded by:

$$D_{a_t}(x^*, x_t) \leq W_t.$$

The evolution of the bound is

$$
\begin{aligned}
\frac{d}{dt} W_t &= \frac{d}{dt} \left( -R_{a_t}(x_t) - \nabla_x R_{a_t}^\top (x^* - x_t) \right) \\
&= -\frac{d}{da} R_{a_t}(x_t) \frac{d}{dt} a_t - \frac{d}{dt} \left( \nabla_x R_{a_t}^\top \right) (x^* - x_t) \\
&\leq \nabla_x f(x_t)^\top (x^* - x_t) \\
&\leq f(x^*) - f(x_t),
\end{aligned}
$$

where again the observations in (13) are used in the first inequality. Therefore the loss converges:

$$
\begin{aligned}
f(x_{T+t}) - f(x^*) &\leq \frac{1}{t} \int_T^{T+t} f(x_s) - f(x^*) ds \\
&\leq \frac{W_T - W_{T+t}}{t} \\
&\leq \frac{W_T}{t} \to 0
\end{aligned}
$$

where the first inequality follows from convexity of the loss and the third inequality from the fact that $W_t \geq D_{a_t}(x^*, x_t) \geq 0$. So the loss converges. We already know from Lemma B.3 that the iterates converge, concluding the convex case.

In case when $f$ satisfies the PL-inequality, we proceed in the same way as Theorem A.3. The evolution of $f$ is given by

$$
\begin{aligned}
df(x_t) &= - \left( \nabla f(x_t)^\top \operatorname{diag} \left( \sqrt{x_t^2 + a_t^2} \right) \nabla f(x_t) + 2\alpha_t \nabla f(x_t)^\top x_t \right) dt \\
&\leq \left( -A_\infty \lambda (f(x_t) - f(x^*)) + \alpha_t C \|x^*\|_{L_2} \right) dt
\end{aligned}
$$

where $C$ is constant depending on the smoothness of $f$ and $A_\infty = 2\min_i u_{0,i} v_{0,i} \exp \left( -2 \int_0^\infty \alpha_s ds \right)$. Then it follows from Gronwall's Lemma that

$$f(x_t) - f(x^*) \leq (f(x_0) - f(x^*)) \exp \left( -A_\infty \lambda t + \int_0^t \alpha_s C \|x^*\|_{L_2} ds \right).$$

It follows from the fact that $\int_0^t \alpha_s ds < \infty$ for all $t \geq 0$ that the loss $f$ convergence. Convergence of the iterates now follows in a similar way as the convex case. $\square$

We now show optimality in the case of underdetermined linear regression Consider a data set $(z_j, y_j)_{i=1}^d$ with $z_j \in \mathbb{R}^n$ and $y_j \in \mathbb{R}$. Let $Z = (z_1, \ldots z_d)$ and $Y = (y_1, \ldots y_d)$. For the regression to be called underdetermined $n > d$.

**Theorem B.2** *In case of under-determined regression consider the loss function $f(x) = \tilde{f}(Zx - Y)$. Assume $f$ satisfies the conditions with at least one of the convergence criteria of Theorem B.1. Then $x_t$ converges to $x^*$ such that*

$$x^* = argmin_{Zx=Y} R_{a_\infty}(x) \tag{14}$$

Proof. Convergence follows from Theorem B.1. It remains to be shown that the optimality conditions of (14) are satisfied. The gradient flow of $R_{a_t}$ satisfies

$$\nabla R_{a_t}(x_t) = Z^\top \int_0^t \nabla \tilde{f}(x_s) ds \in \operatorname{span}\{Z^\top\}.$$

This quantity is well defined for all $t \geq 0$ because $\nabla \tilde{f}$ is locally Lipschitz (because $f$ has to be locally Lipschitz). Therefore taking the $t \to \infty$ yields the KKT conditions of the optimization problem in (14). $\square$

### B.1 DISCUSSION OF THE PROOF

Most of the proof follows the same arguments as in (Alvarez et al., 2004; Li et al., 2022; Pesme et al., 2021). The notable differences are showing that the loss becomes decreasing over time and the observations made in (13).

## C DETAILS EXPERIMENTS

In this section, we provide the details of the experiments. In addition, there are additional figures given.

**Compute**  The codebase for the experiments is written in PyTorch and torchvision and their relevant primitives for model construction and data-related operations. The experiments in the paper are trained on an NVIDIA A6000. In addition, the diagonal linear network is trained on a CPU 13th Gen INTEL(R) Core(TM) i9-13900H.

### C.1 DIAGONAL LINEAR NETWORK

For each setting, different regularization schemes are tried. In total, 7 options are tried. 4 of the schedules are constant i.e. the regularization stays the same during training. The remaining 3 are decaying schedules. These schedules we name harmonic, quadratic, and geometric. The schedules are described by the following recurrent relations:

$$ h_k = \frac{1}{k} \qquad , \qquad q_k = \frac{1}{k^2} \qquad \text{and} \qquad g_k = p^k $$

where we have set $p = 0.95$. These schedules lead to a total strength of regularization applied. We denote $A := \int_0^t \alpha_s ds$ the total strength of the regularization. So in practice, it is the weighted sum of the regularization strength.

In Figure 5 we present the trajectories of all the schedules for the 3 considered settings. We observe that our regularization performs the best with the decaying schedules as predicted by the theory. The other methods need constant regularization to perform well as already mentioned in Remark 2.1. Note that, PILoT also can perform well with constant regularization (see Figure 5.

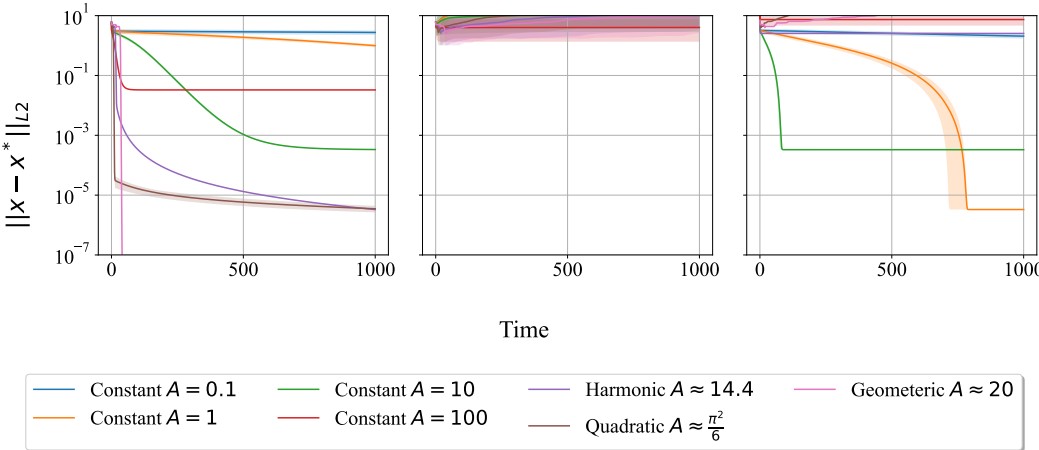

Figure 5: All runs for the diagonal linear network. From left to right $m \odot w$ with PILoT initalization, $m \odot w$ with spred initialization, and $x$ with $L_1$ regularization

### C.2 ONE-SHOT

In this section we give the additional details for the one-shot experiments. In Table 2 the hyperparameters for the CIFAR 10 and 100 experiments are given. To determine which configuration is best

for which sparsity level we compute the validation accuracy at multiple levels and choose the level just before the accuracy drops $1\%$ or in the high-sparsity regime $2\%$. Moreover, we use $s = -200$ for STR.

For the ImageNet experiment we use the setup of (Kusupati et al., 2020). For PILoT and spred we in addition use $L_2$ regularization to compensate for the weight decay i.e. we add a term $(m \odot w)^2$ with strength $0.000030517578125/2$, which is based on the weight decay strength in (Kusupati et al., 2020) for the baseline. Furthermore, weight decay is turned off for the other parameters. In Table 3 we present the configurations that correspond to the values in Table 1. Note for all PILoT configs $\delta = 1.01$ is used.

Table 2: One-shot experiment

| Parameter | Setting | Comments |
|---|---|---|
| Optimizer | SGD | |
| Momentum | 0.9 | |
| Batch size | 256 | |
| Activation function | ReLu | |
| Weight decay[2] | $10^{-4}$ | |
| Base learning rate | $\{0.1, 0.2\}$ | |
| Epochs | 150 | |
| Warmup period | 0 | |
| Initialization | Kaiming normal | |
| Scaling | 1 | Only for $m \odot w$ |
| $\delta$ | 1.01 | |
| $K$ | 8000 | |
| **CIFAR 10** | | |
| Learning rate schedule | cosine warmup | |
| **CIFAR 100** | | |
| Learning rate schedule | step warmup | |

Table 3: ResNet-50 on ImageNet configurations for each sparsity (%).

| Method | $\alpha_{init}$ | $K$ | sparsity |
|---|---|---|---|
| spred[3] | $2e-5$ | - | 80.00 |
| spred | $3e-6$ | - | 79.03 |
| PILoT | $7e-6$ | 60000 | 80.00 |
| spred | $5e-6$ | - | 89.26 |
| PILoT | $1e-5$ | 60000 | 88.00 |
| PILoT | $1.4e-5$ | 60000 | 91.00 |

| Method | $\alpha_{init}$ | $K$ | sparsity |
|---|---|---|---|
| spred | $2e-5$ | - | 94.50 |
| PILoT | $2e-5$ | 60000 | 94.00 |
| PILoT | $3e-5$ | 60000 | 95.00 |
| PILoT | $3e-5$ | 60000 | 95.60 |
| PILoT | $3e-5$ | 60000 | 96.00 |
| spred | $3e-5$ | - | 97.19 |
| PILoT | $4e-5$ | 40000 | 97.19 |
| spred | $5e-5$ | - | 98.20 |
| PILoT | $5e-5$ | 20000 | 97.75 |
| PILoT | $7e-5$ | 20000 | 98.20 |

**Label smoothing** Altough PILoT is competitive in the $80\% - 90\%$ sparsity range it is not SOTA. Nevertheless, if we turn of label smoothing in the experiment PILoT outperforms STR in this range as well. The only change for STR is turning of labelsmoothing. For PILoT we use two different configurations. We use $L_2$ regularization i.e. $\|m \odot w\|_{L_2}^2$ regularization set to $5 \cdot 10^{-5}$ instead of the value from STR experiment. We use $K = 600000$ and $\delta = 1.01$. Furthermore, the strength of the PILoT regularization is initialized at $\{1 \cdot 10^{-5}, 2 \cdot 10^{-5}\}$ and no weight decay is used on the rest of the parameters. The results are given in Table C.2.

---

[2] Applied to the other parameters
[3] Starting from a pretrained model with $77\%$ validation accuracy

Table 4: Extra experiment ResNet-50 on ImageNet sparsity (%) versus accuracy (%) without label smoothing.

| Method | Top-1 Acc | Sparsity |
|--------|-----------|----------|
| ResNet-50 | 75.80 | 0 |
| STR | 73.03 | **79.03** |
| PILoT | **74.72** | **79.03** |
| STR | 71.6 | 89.26 |
| PILoT | **73.21** | **91.41** |

### C.3 ITTERATIVE PRUNING

In Table 5 the details of the ImageNet the experiment are given. Note the base learning rate $0.1$ is for the baseline and $0.2$ is used for our parameterization combined with scaling $1$. In addition, the $L_2$ regularization denotes the reparameterization of the original weight decay. Thus for PILoT, in this case, we use that instead. Moreover, all runs have been done for $3$ different seeds. Furthermore, we provide additional experiments on CIFAR 10 and 100 with ResNet-20 and ResNet-18 respectively in Figure 6. The details are given in Table 6

Table 5: WR and LRR experiment on ImageNet

| Parameter | Setting | Comments |
|-----------|---------|----------|
| Optimizer | SGD | |
| Momentum | 0.9 | |
| Batch size | 512 | |
| Activation function | ReLu | |
| Weight decay | $\{0, 10^{-4}\}$ | |
| Learning rate schedule | step warmup | |
| Base learning rate | $\{0.1, 0.2\}$ | |
| Cycles | 25 | |
| Pruning rate | 0.8 | |
| Epochs per cycle | 90 | |
| Warmup period | 10 | |
| Initialization | Kaiming normal | |
| $L_2$ regularization | $5 \cdot 10^{-5}$ | Only for $m \odot w$ |
| PILOT regularization | $\{0\}$ | Only for $m \odot w$ |
| Scaling | 1 | Only for $m \odot w$ |
| $\delta$ | 1 | Only for $m \odot w$ |
| $K$ | $-$ | Only for $m \odot w$ |

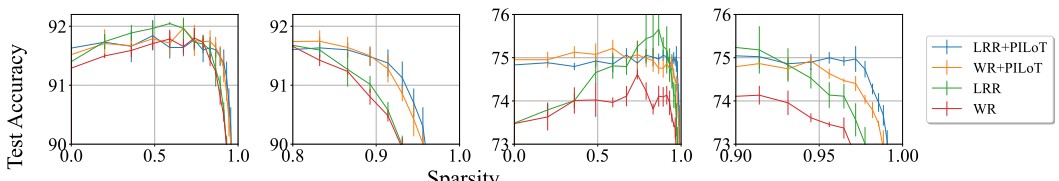

Figure 6: Learning Rate Rewinding (LRR) and Weight Rewinding (WR) with PILoT shows improvement over the baseline itterative pruning methods for CIFAR 10 and 100.

## D REMARK ON NEURONWISE PRUNING

In the main text, we have used mirror flow to describe the implicit bias of parameterwise pruning. In this section, we show that neuronwise pruning can not be analyzed in the same way. We first define neuronwise pruning. Next, we paraphrase the necessary condition such that the implicit bias can be

Table 6: WR and LRR experiment on CIFAR 10 and 100

| Parameter | Setting | Comments |
|---|---|---|
| Optimizer | SGD | |
| Momentum | 0.9 | |
| Batch size | 256 | |
| Activation function | ReLu | |
| Weight decay | $10^{-4}$ | |
| Learning rate schedule | step warmup | |
| Base learning rate | $\{0.1, 0.2\}$ | |
| Cycles | 25 | |
| Pruning rate | 0.8 | |
| Epochs per cycle | 150 | |
| Warmup period | 50 | |
| Initialization | Kaiming normal | |
| $L_2$ regularization | 0 | Only for $m \odot w$ |
| PILOT regularization | $\{10^{-4}\}$ | Only for $m \odot w$ |
| Scaling | 1 | Only for $m \odot w$ |
| $\delta$ | 1 | Only for $m \odot w$ |
| $K$ | − | Only for $m \odot w$ |

described by a mirror flow from (Li et al., 2022). Finally, we show neuronwise pruning violates this condition pointing out a limitation of the framework.

Consider the parameterization for a function with $p$ neurons, $g : \mathbb{R}^p \times \mathbb{R}^{n_1} \times \ldots \times \mathbb{R}^{n_p}$,

$$g(m, w_1, \ldots, w_p) = (m_1 w_1, \ldots, m_p w_p)$$

where $m_i \in \mathbb{R}$ is the mask and $w_i \in \mathbb{R}^{n_i}$ are the neurons.

We state the necessary condition for a parameterization to induce a mirror flow.

**Theorem D.1** *(Theorem 4.10 (Li et al., 2022)) The Lie bracket span of $\{\nabla_i g\}_{i=1}^n$ is in the kernel of Jacobian $\partial g$.*

Now, we use this theorem to show that neuronwise pruning does not induce a mirror flow.

**Lemma D.1** *Neuronwise pruning violates Theorem D.1.*

Proof. We show that for $p = 1$ the condition is already violated. This implies that in the general case, the condition is also violated as the neurons themselves are commuting with each other as they are parameterized separate.

To see this we can explicitly check the commuting condition for the following parameterization $g : \mathbb{R} \times \mathbb{R}^n \to \mathbb{R}^n$

$$g(m, w) = mw$$

Then the gradients (Jacobians) and Hessian's are given by:

$$\nabla g_i = \begin{pmatrix} w_i \\ m\mathbb{I}_{i=1} \\ \vdots \\ m\mathbb{I}_{i=n_1} \end{pmatrix} \quad \text{and} \quad Hg_i = \begin{pmatrix} 0 & \mathbb{I}_{i=1} & \ldots & \mathbb{I}_{i=n} \\ \mathbb{I}_{i=1} & 0 & \ldots & 0 \\ \vdots & & & \vdots \\ \mathbb{I}_{i=n} & 0 & \ldots & 0 \end{pmatrix} \quad \text{for } i = 1, 2$$

Computing $Hg_i \nabla g_j$

$$Hg_i \nabla g_j = w_j \begin{pmatrix} 0 \\ \mathbb{I}_{i=1} \\ \vdots \\ \mathbb{I}_{i=n} \end{pmatrix}$$

we compute the Lie brackets which span a subspace of the Lie Algebra $LIE^{\geq 2}(\partial g)$. The subspace is spanned by

$$\text{span}\left( w_j \begin{pmatrix} 0 \\ \mathbb{I}_{i=1} \\ \vdots \\ \mathbb{I}_{i=n} \end{pmatrix} - w_i \begin{pmatrix} 0 \\ \mathbb{I}_{j=1} \\ \vdots \\ \mathbb{I}_{j=n} \end{pmatrix} \text{ for } i,j \in [n] \right) \subset LIE^{\geq 2}(\partial g).$$

Clearly, this span is not in $Ker(\partial g)$ as can be shown by a direct computation:

$$(\partial g)\left( w_j \begin{pmatrix} 0 \\ \mathbb{I}_{i=1} \\ \vdots \\ \mathbb{I}_{i=n} \end{pmatrix} - w_i \begin{pmatrix} 0 \\ \mathbb{I}_{j=1} \\ \vdots \\ \mathbb{I}_{j=n} \end{pmatrix} \right) = (\mathbb{I}_{i=1}mw_j - \mathbb{I}_{j=1}mw_i, \ldots, \mathbb{I}_{i=n}mw_i - \mathbb{I}_{j=1}mw_i)$$

For the span to be in the kernel we need either that all $w_i = 0$ for all $i \in [n]$ or $m = 0$. In both cases this implies that $g(m,w) \in \{0\}$. This implies that a mirror flow is only well-defined if the parameterization is zero. Hence, neuron-wise continuous sparsification does not induce a mirror flow. $\square$

