# OpenReview forum: "Mask in the Mirror: Implicit Sparsification"
_ICLR.cc/2025/Conference — ICLR 2025 Poster_

### Official Review · Reviewer_UaBp · 2024-10-31

**Soundness:** 3
**Presentation:** 3
**Contribution:** 3
**Rating:** 6
**Confidence:** 3

**Summary:**

This paper studies the implicit sparse regularization of continuous specification. By studying the implicit bias of a specific mirror flow with a time-dependent Bregman potential, the authors propose a novel continuous sparsification method  "parametric implicit lottery ticket" (PILoT). The convergence and optimality is shown theoretically, and the effectiveness in real-world examples is demonstrated numerically.

**Strengths:**

The paper is well-written and easy to follow. I didn't go through the proofs, but the illustrations make sense, and the technical contribution is solid.

**Weaknesses:**

One point that might limit the significance of the conclusion is that the implicit regularization appears after the model fits the data perfectly (Theorem 2.3), which might not hold in reality.

**Questions:**

The literature below is related but missed

* Vaskevicius et al, Implicit regularization for optimal sparse recovery, NeurIPS 2019.
* Li et al, Implicit Sparse Regularization: The Impact of Depth and Early Stopping, NeurIPS 2021.
* Zhao et al, High-Dimensional Linear Regression via Implicit Regularization, Biometrika 2022.
* Li et al, Implicit bias of gradient descent on reparametrized models: On equivalence to mirror descent, NeurIPS 2022.
* Li et al, Implicit Regularization for Group Sparsity, ICLR 2023.

Minor comments:
* .(dot) is missed for the leading bold text in Section 1.1
* Line 213, "(Ziyin & Wang, 2023)..." is not a sentence.
* Line 404, "return..." goes to the next line

---

> ### Author Response · Authors · 2024-11-20
>
> We would like to express our gratitude for the reviewer’s time and effort in providing valuable comments on our manuscript and appreciate the acknowledgment of our theoretical and practical contributions. We have revised our manuscript accordingly and provide a point by point response below. A latexdiff comparison of our old and revised draft can be found in the zip folder of the submitted supplementary material. (The file is named PILoTlatexdiff.pdf.)
>
> __Optimality concern:__
>
> We acknowledge the concern that, in practice, perfectly fitting the data may not always be achievable. Yet, even if this is not achieved, the combined effects of the implicit bias and explicit regularization remain significant. As shown by Eq.~(6), the training dynamics smoothly transitions the implicit bias from $L_2$ to $L_1$. This transition is the main reason for the SOTA performance of PILoT.
>
> __Related literature:__
>
> Thank you for highlighting the relevant literature on implicit bias.
> We have incorporated the suggested references into the implicit bias section of the literature review.
>
> __Minor comments:__
>
> Moreover, the minor comments have been implemented.

---

> > ### Comment · Reviewer_UaBp · 2024-11-29
> >
> > Thanks for the reply. Maybe I should have made it clear: regarding the literature I mentioned, I was expecting more discussions of this work relating to neural network structured/unstructured sparsification under realistic/perfect fitting scenarios to reveal the significance of this work. For example, (Vaskevicius 2019, Li 2021, and Zhao 2022) are the early works on neural network sparsification in a realistic setting. I also found one more paper (Dai 2021), along with (Li 2022, and Li 2023) explores various structured sparse biases of neural networks. I thought such discussions would help position the contribution of this work in a better place.
> >
> > I will keep my rating unchanged and lean towards acceptance.
> >
> > * Dai et al, Representation Costs of Linear Neural Networks: Analysis and Design. NeurIPS 2021.

---

> > > ### Author Response · Authors · 2024-12-02
> > >
> > > Thank you for your valuable suggestion and leaning towards acceptance.
> > >
> > > We would be happy to extend our literature discussion in the following. In contrast to the suggested work, we highlight the effect of initialization, explicit regularization, and how they impact the implicit bias for general non-convex optimization problems. Most importantly, we propose to exploit this insight for the dynamic control of this bias.
> > >
> > > While the scenarios discussed in Vaskevicius et al. (2019), Li et al. (2021), and Zhao et al. (2022) consider discrete gradient updates, their scope is limited as they focus solely on linear regression without explicit regularization. This limitation makes the analysis more tractable but does not directly extend to the broader context of neural network sparsification (which we study). However, the non-convex nature of general neural network sparsification requires the more adaptable mirror flow framework. Within this framework, we can isolate and study the effect of reparameterization combined with weight decay, as presented in Eq. (6). We will emphasize this distinction relative to the discrete case on Line 173.
> > >
> > > In the context of gradient flow, Li et al. (2022) provide necessary and sufficient conditions for parameterizations that can be described by a mirror flow for non-convex objectives. As outlined in the introduction and related work, we extend this framework for the specific parameterization $m \odot w $  to include weight decay, which induces a time-dependent mirror flow.
> > >
> > > Furthermore, we highlight similar limitations for neuron-wise sparsification (in Appendix D). Li et al. (2023) demonstrate that group sparsity dynamics cannot be captured by a mirror flow, as the parameterization violates the necessary conditions established in Li et al. (2022). We will incorporate this connection in Line 147.
> > >
> > > Moreover, Dai et al. (2021) examine the influence of constant explicit regularization on model complexity across various neural network architectures - but not learning dynamics. For instance, in the case of linear regression with overparameterization $m \odot w$ and weight decay, the optimization problem is equivalent to LASSO. This equivalence serves as a precursor to the analysis by Ziyin and Wang (2023) of the spred algorithm, which extends this relationship to non-convex objectives. We will add this connection on Line 167. Our work, however, highlights that $m \odot w$ and weight decay are not equivalent to LASSO but identify crucial differences in the learning dynamics, which suggest early implicit L2 regularization that transitions into implicit L1 regularization over time.

---

### Official Review · Reviewer_jfBc · 2024-11-04

**Soundness:** 2
**Presentation:** 2
**Contribution:** 2
**Rating:** 6
**Confidence:** 3

**Summary:**

In this paper, the authors focus on the problem of finding sparse neural networks. They propose a new method called PILoT based on theoretical analysis on simple diagonal linear networks. They use the insights gained from theoretical analysis to improve previous method called spred to introduce dynamic regularization strength and initialization scheme. Experiments are provided to show the improvement of the proposed method.

**Strengths:**

-	Studying the sparsification of neural networks is an interesting research problem.
-	The proposed method that applies the idea of implicit/explicit regularization to pruning seems to be new.
-	The performance of the proposed method shown in experiments suggests that there is some improvement over the previous works.

**Weaknesses:**

-	Many places are not very clear to me as a reader.
-	See questions section below.

**Questions:**

-	In line 228, I was wondering why ‘’the potential $R$ attains its global minimum at the initialization $x_0$... In consequence, we would not promote actual sparsity’’. I’m confused about this. As many paper that authors cited, when initialization scale goes to 0, the final implicit regularization is $\ell_1$ norm that promotes the sparsity.
-	In Theorem 2.2, what is $\alpha_\infty$? Is it regularization $\alpha_t$ at time $t\to\infty$? If not, what does it mean? If so, it seems to me that $\alpha_t\to 0$ given the assumptions. This would make $\alpha_\infty=0$, which makes the statement very weird. Moreover, if $\alpha_\infty\neq 0$, then why $x_t$ would converge to a minimizer of $f$ given the existence of a regularization term.
-	In line 347, I wonder if authors could explain more on why eq (6) explains spred performs better than LASSO. I believe there are many classical algorithms that can solve LASSO better than just using gradient descent/gradient flow, and they may not have such problems.
-	In line 374, I wonder why $\beta=1$ is motivated by the discretization of the gradient flow. It is not clear to me what are the connections here.
-	In the proposed algorithm PILoT (Algorithm 1), I wonder why $\alpha_k$ should be increased when the sparsity $\|m_k \odot w_k \|_1\ge K$. Since decreasing $\alpha_k$ would promotes better sparsity, I feel $\alpha_k$ should be decreased in this case?
-	In experiments, what are the schedules of $\alpha_k$ that are used?


Typo:

-	Below eq (6), missing ( in expression of $a_t$.

---

> ### Author Response · Authors · 2024-11-20
>
> We would like to express our gratitude for the reviewer’s time and effort in providing valuable comments on our manuscript and appreciate the acknowledgment of our theoretical and practical contributions. We have revised our manuscript accordingly and provide a point by point response below. A latexdiff comparison of our old and revised draft can be found in the zip folder of the submitted supplementary material. (The file is named PILoTlatexdiff.pdf.)
>
> __Writing quality:__
>
> We have revised our manuscript and believe that we have improved the clarity of the presentation substantially. Our changes are highlighted in the file PILoTlatexdiff.pdf in the zip folder of the supplementary material (using latexdiff). In case of any remaining concerns, we would be happy to resolve them upon request.
>
> In summary:
> 1) We have streamlined the introduction to clearly reflect the following logical flow:
> - We motivate continuous sparsification and that its implicit bias towards sparsity forms the basis of the method spred (which relies on the m w parameterization and constant weight decay).
> - We argue that the fact that the bias is implicit allows spread to outperform LASSO (and thus explicit $ L_1$ regularization), as it induces different learning dynamics.
> - Our analysis of the learning dynamics further extends the implicit bias framework by a time-dependent explicit regularization that induces a smooth transition from $ L_2$ to $ L_1$ regularization.
> - As we show, this time-dependent regularization can resolve two main caveats of spred (related to its initialization and constant regularization). These caveats can prevent it from entering the so-called rich regime and thus converge to a good solution.
> - This motivates our method PILoT, which improves the initialization, proposes a time-dependent regularization, and outperforms state-of-the-art baselines (including spred) in experiments.
> 2) We have added explanations in the theory sections (Sections 2 and 3) and ensured that all notation and concepts are defined the first time they are introduced.
> 3) Our conclusions follow the same storyline as the introduction, only in condensed form with a focus on our method PILoT.

---

> > ### Author Response · Authors · 2024-11-20
> >
> > __Q1. Global minimum of R:__
> >
> > The cited papers initialize network parameters at zero. However, this approach is unsuitable for deep neural networks because it prevents any signal from propagating through the initial function. To address this, we aim to use standard neural network initializations, which requires resolving this issue. To emphasize its importance, we explicitly identify this challenge as one of two key limitations of the current mirror flow framework in the manuscript.
> >
> > **Q2. $\alpha_{\infty}$ versus $a_{\infty}$ in Thm. 2.2:**
> >
> > Theorem 2.2 does not use $\alpha_{\infty}$, but it does involve $a_{\infty}$. Indeed, $\alpha_t \rightarrow 0$, but  $a_{\infty} >0$ remains positive. However, $a_{\infty} >0$ can be exponentially small, which may lead to slow convergence as discussed in Lines 306-312.
> > This issue can be mitigated under certain conditions, as highlighted in Remark 2.2.
> >
> > __Q3: The performance of spred and classical LASSO solvers:__
> >
> > As shown in Eq. (6), spred exponentially reduces redundant features, as discussed in Lines 372–377. This explains its superior performance compared to gradient descent/flow-based LASSO methods, as observed in Figure 3. While we acknowledge that there exist more advanced LASSO solvers, such as LARS LASSO (Hastie et al. 2004) and FISTA (Beck & Teboulle 2009), these methods have also limitations. For instance, Figure 7 in the appendix of (Ziyin & Wang 2022) demonstrates that both LASSO and LARS LASSO require significantly more compute time as the number of features increases, with no noticeable improvement in accuracy. This explains why they are not applied in the context of large deep neural networks. Spred’s ability to exponentially reduce redundant features is responsible for its lower computational cost.
> > Nevertheless, FISTA can perform on par with PiLOT and outperform spred in sparse linear regression due to spred's inability to flip signs. However, it is not surprising that specialized algorithms perform well on linear regression. While we are interested in methods that scale to deep neural networks, the diagonal linear network example serves mostly as an analytically tractable testbed and illustration. It is sufficient to highlight limitations of spred, for instance.
> > Note that both STR and FISTA rely on soft thresholding, where STR is tailored to neural network pruning. As our deep learning experiments demonstrate, PILoT outperforms STR in the high-sparsity regime.
> >
> > Efron, B., Hastie, T., Johnstone, I., & Tibshirani, R. (2004). Least angle regression. Annals of Statistics, 32, 407-499.
> >
> > Beck, A., & Teboulle, M. (2009). A Fast Iterative Shrinkage-Thresholding Algorithm for Linear Inverse Problems. SIAM J. Imaging Sci., 2, 183-202.
> >
> > Ziyin, L., & Wang, Z. (2022). spred: Solving L1 Penalty with SGD. International Conference on Machine Learning.
> >
> > __Q4. Discretization of gradient flow:__
> >
> > The rationale for selecting $\beta$ is best explained with the help of Eq. (6).
> > At initialization, the gradient is scaled by the coefficient $\eta \sqrt{x_0^2 + \beta^2}$, where $\eta >0$ is the learning rate.
> > When $x_0 = 0$, the effective learning rate becomes $ \eta |\beta|$.
> > By setting $\beta = 1$ the effective learning rate at zero simplifies to $\eta$.
> > We have clarified this point explicitly in Line 379: “After discretizing Eq. (6), the effective learning rate at initialization $x_0 = 0$ is $\eta |\beta|$, where $\eta >0$ is the learning rate.
> > Therefore, we use $\beta = 1$ in the experiments so that the learning rate $\eta$ is not altered.”
> >
> >
> > __Q5: Criteria for increasing $\alpha_k$:__
> >
> > Please note that increasing $\alpha_k$ promotes sparsity,  while decreasing it enhances accuracy. A larger $\alpha_k$  leads to a smaller $a_t$, shifting the implicit bias towards $L_1$, as illustrated in Figure 1.
> > To clarify, we have added the following explanation at Line 392: “The regularization strength (and thus sparsity) grows if the sparsity threshold has not been reached yet and the training accuracy has increased in the previous gradient update step.”
> >
> > __Q6. Schedules of $\alpha_k$:__
> >
> > The schedules for $\alpha_k$ follow the procedure of Algorithm 1 for both the one-shot and iterative pruning experiments. For the diagonal linear network experiment, we tested various schedules and reported the best-performing ones for each method in Figure 2. Details of these schedules are provided in Appendix C.1 for PILoT. Notably, the geometric schedule seems to be the best, serving as additional motivation for our design of Algorithm 1. Moreover, we mention this now explicitly in the manuscript at Line 442.

---

> ### Comment · Reviewer_jfBc · 2024-11-25
>
> I thank authors for providing response to address my concerns. I have carefully reviewed the response as well as the comments from other reviewers. I would recommend making $a_\infty$ clear in Theorem 2.2, as $a_t$ seems not formally defined before this point and $a_\infty$ looks like $\alpha_\infty$, which may make people confusing. I will increase my score.

---

> > ### Author Response · Authors · 2024-11-25
> >
> > Thank you for the valuable suggestion and for increasing your score. Accordingly, we have renamed $a_{\infty}$ to $A_{\infty}$ and have emphasized the connection: $A_{\infty} = \min_i \left(m_{0,i}^2 - w_{0,i}^2\right) \exp \left(-2\int_0^{\infty} \alpha_s ds\right)$.

---

### Official Review · Reviewer_L22t · 2024-11-08

**Soundness:** 3
**Presentation:** 1
**Contribution:** 3
**Rating:** 6
**Confidence:** 3

**Summary:**

This paper studies the mirror flow of the time-dependent $L_2$-regularized optimization problem which is parametrized by masked weights $x=m\odot w$. The authors derive the corresponding Bregman potential for the mirror flow and prove a convergence result for quasi-convex loss functions. Inspired by the forms of the derived Bregman potential and the gradient flow, the authors propose PILoT, which takes (i) a strategy to control the strength of $L_2$-regularization dynamically and (ii) a initialization for $m$ and $w$ that enables sign flipping. Evaluations show that PILoT outperforms baselines on CIFAR 10 and CIFAR 100.

**Strengths:**

* The theoretical result seems sound though I didn't check the proofs.
* The Bregman potential offers insight on $L_1$ regularization effect.
* The gradient flow also offers insights on why spred outperforms Lasso from the perspective of convergence rate.
* I appreciate that the authors also develop an algorithm inspired by the theory.

**Weaknesses:**

The writing is poor. Some notations are never explained, e.g., $m^2$ (though I can see that's the square of $L_2$ norm, but it is not the standard notation). The experiments are also a bit hard to read. Please see questions.

**Questions:**

1. I found table 1 confusing. Why do all methods have duplicates in the table?
2. At line 227, why does $R$ attain its global minimum at $x_0$ when $R$ is $L_1$ norm?
3. At line 218, is there any formal result stating that explicit regularization would lead to a trade-off? I can see that implicit bias of the reparametrized optimization is helping achieve both sparsification and low loss, but does that imply an explicit regularization forces a trade-off?

---

> ### Author Response · Authors · 2024-11-20
>
> We would like to express our gratitude for the reviewer’s time and effort in providing valuable comments on our manuscript and appreciate the acknowledgment of our theoretical and practical contributions. We have revised our manuscript accordingly and provide a point by point response below. A latexdiff comparison of our old and revised draft can be found in the zip folder of the submitted supplementary material. (The file is named PILoTlatexdiff.pdf.)
>
> __Writing quality:__
>
> We have revised our manuscript and believe that we have improved the clarity of the presentation substantially. In case of any remaining concerns, we would be happy to resolve them upon request. Our changes are highlighted in the file PILoTlatexdiff.pdf in the zip folder of the supplementary material (using latexdiff).
>
> In summary:
> 1) We have streamlined the introduction to clearly reflect the following logical flow:
> - We motivate continuous sparsification and that its implicit bias towards sparsity forms the basis of the method spred (which relies on the m w parameterization and constant weight decay).
> - We argue that the fact that the bias is implicit allows spread to outperform LASSO (and thus explicit $ L_1$ regularization), as it induces different learning dynamics.
> - Our analysis of the learning dynamics further extends the implicit bias framework by a time-dependent explicit regularization that induces a smooth transition from $ L_2$ to $ L_1$ regularization.
> - As we show, this time-dependent regularization can resolve two main caveats of spred (related to its initialization and constant regularization). These caveats can prevent it from entering the so-called rich regime and thus converge to a good solution.
> - This motivates our method PILoT, which improves the initialization, proposes a time-dependent regularization, and outperforms state-of-the-art baselines (including spred) in experiments.
> 2) We have added explanations in the theory sections (Sections 2 and 3) and ensured that all notation and concepts are defined the first time they are introduced.
> 3) Our conclusions follow the same storyline as the introduction, only in condensed form with a focus on our method PILoT.
>
> __Q1. Duplicates in tables:__
>
> Continuous sparsification poses a challenge in achieving a given sparsity target, which we need for a fair comparison of different methods. As the exact same sparsity target is not well reachable, we present several results that attain very close sparsity values (which are slightly higher and lower than the sparsity target.)
> The hyperparameter configurations used to obtain the reported results for spred and PILoT are detailed in Appendix C, Table 3.
> In addition, the entries for STR are directly taken from Table 1 (Kusupati et al., 2020), which also uses multiple configurations. This ensures a consistent and fair evaluation across methods.
>
> Aditya Kusupati, Vivek Ramanujan, Raghav Somani, Mitchell Wortsman, Prateek Jain, Sham
> Kakade, and Ali Farhadi. Soft threshold weight reparameterization for learnable sparsity. In
> Proceedings of the International Conference on Machine Learning, July 2020.
>
> __Q2. Global minimum of $R$:__
>
> We have added a detailed explanation in the text for clarity, as this point relates to one of our main contributions. The mirror flow framework suggests that the implicit bias promotes solutions that are close to the initialization $x_0$, since $R$ attains its global minimum there. Consequently, the implicit bias does not exactly correspond to $L_1$ when $x_0 \neq 0$. For that reason, initializations close to $0$ are preferred to implicitly regularize towards sparsity. We overcome this caveat by making the regularization time-dependent. Figure 1 illustrates how the global minimum moves towards zero when the total regularization $\int_0^t \alpha_s ds$ increases.
>
> __Q3. Trade-off:__
>
> A well-known theoretical result, the bias-variance trade-off, states that increasing explicit regularization reduces the variance at the cost of increased bias, potentially leading  to worse prediction.
> For a detailed treatment of the trade-off, see Chapter 6 of (Witten et al. 2013).
> In addition, an optimization-based explanation  for this trade-off is that explicit regularization introduces a second optimization objective. The fixed strength of the explicit regularization determines the relative importance of the two objectives.
> In contrast, the implicit bias or mirror flow formulation helps mitigate this trade-off, by reducing the strength of explicit regularization over time. The mitigation of the trade-off is best illustrated by Theorem 2.3 where we recover the unbiased prediction while still minimizing the Bregman potentia, effectively reducing variance without incurring significant bias.
>
> James, G.M., Witten, D.M., Hastie, T.J., & Tibshirani, R. (2013). An Introduction to Statistical Learning. Springer Texts in Statistics.

---

> > ### Comment · Reviewer_L22t · 2024-11-26
> >
> > Thank you for your response. I took a pass at the revision and it is improved on writing. I will increase my rating to 6.

---

### Official Review · Reviewer_cfhK · 2024-11-08

**Soundness:** 3
**Presentation:** 1
**Contribution:** 2
**Rating:** 5
**Confidence:** 3

**Summary:**

"Spred" is a recent algorithm that implicitly achieves sparse solutions through reparameterization. To understand why "Spred" outperforms LASSO, the authors analyzed its training dynamics using the mirror flow framework. They demonstrated that when the regularization term in "Spred" is time-dependent, it can smoothly transition between implicit L2 and L1 regularization. Building on this insight, they introduced PILoT, an approach that dynamically adjusts the regularization strength. Finally, the authors validated the effectiveness of PILoT through experiments on CIFAR-10/100 and ImageNet datasets.

**Strengths:**

The theoretical analysis of the reparametrization with time-varying weight decay is solid and novel. It is valuable to first develop a deep theoretical understanding, which then serves as a basis for algorithmic improvements.

**Weaknesses:**

**Major:**
- **Inaccurate and unclear writing**: The presentation in the paper could benefit from a substantial revision to improve clarity, particularly in the latter half of Section 1 and Sections 2, 3, and 5. Additionally, some inaccuracies need to be addressed. For example, the abstract states that "A key factor in their (continuous sparsification) success is the implicit L1 regularization induced by jointly learning both mask and weight variables." However, the discussion from Lines 54-65 suggests that this implicit L1 regularization is exclusive to the "Spred" algorithm, which utilizes reparameterization. The paper is currently difficult to follow, which is my primary concern.
- **Computation and memory costs of reparameterization**: The reparameterization approach proposed in the algorithm effectively doubles the number of parameters, resulting in increased computational and memory demands. However, this is not discussed in the paper.

**Minor:**
- **Figure 4's legend is confusing**: It would be clearer to rename "mw LRR" to "LRR with reparameterization" and "x LRR" to "LRR."
- **Unclear variables in Algorithm 1**: The variables "u_0" and "v_0" introduced in Algorithm 1 are not referred to in subsequent sections, leading to confusion about their purpose.

**Questions:**

N/A

---

> ### Author Response · Authors · 2024-11-20
>
> We would like to express our gratitude for the reviewer’s time and effort in providing valuable comments on our manuscript and appreciate the acknowledgment of our theoretical and practical contributions. We have revised our manuscript accordingly and provide a point by point response below. A latexdiff comparison of our old and revised draft can be found in the zip folder of the submitted supplementary material. (The file is named PILoTlatexdiff.pdf.)
>
>
> __Inaccurate and unclear writing:__
>
> We have revised our manuscript and believe that we have improved the clarity of the presentation substantially. In case of any remaining concerns, we would be happy to resolve them upon request. Our changes are highlighted in the file PILoTlatexdiff.pdf in the zip folder of the supplementary material using latexdiff).
>
> In particular, we have resolved the pointed out ambiguity. Any continuous sparsification, which resembles the parameterization mw, induces the analyzed implicit bias. Lines 100-102 now explicitly state this insight.
>
> In summary:
> 1) We have streamlined the introduction to clearly reflect the following logical flow:
> - We motivate continuous sparsification and that its implicit bias towards sparsity forms the basis of the method spred (which relies on the m w parameterization and constant weight decay).
> - We argue that the fact that the bias is implicit allows spread to outperform LASSO (and thus explicit $ L_1$ regularization), as it induces different learning dynamics.
> - Our analysis of the learning dynamics further extends the implicit bias framework by a time-dependent explicit regularization that induces a smooth transition from $ L_2$ to $ L_1$ regularization.
> - As we show, this time-dependent regularization can resolve two main caveats of spred (related to its initialization and constant regularization). These caveats can prevent it from entering the so-called rich regime and thus converge to a good solution.
> - This motivates our method PILoT, which improves the initialization, proposes a time-dependent regularization, and outperforms state-of-the-art baselines (including spred) in experiments.
> 2) We have added explanations in the theory sections (Sections 2 and 3) and ensured that all notation and concepts are defined the first time they are introduced.
> 3) Our conclusions follow the same storyline as the introduction, only in condensed form with a focus on our method PILoT.
>
>
> __Computation and memory costs of reparameterization:__
>
> To address this concern, we have added the following paragraph in Lines 428-432:
> “As most other continuous sparsification approaches, note that PILoT doubles the number of parameters during training. Yet, according to Ziyin & Wang (2023), the training time of a ResNet50 with $m \odot w$ parameterization on ImageNet increases roughly by 5% only and the memory cost is negligible if the batch size is larger than 50. At inference, we would return to the original representation x and therefore benefit from the improved sparsification.”
>
>
> __Minor:__
>
> - In Figure 4, we have renamed mw LRR to LRR+PILoT and x LRR to LRR to improve clarity.
> -  In Algorithm 1, we have removed the variables $u_0$ and $v_0$. The initialization notation ($m_0$ and $w_0$) is defined in Section 2.

---

> > ### Author Response · Authors · 2024-12-03
> >
> > We would like to thank you for your valuable review. We believe that we have addressed all points of criticism but would be happy to address any open issues if there should remain any. Since the discussion period is approaching its end, we would highly appreciate your feedback.

---

### Meta-Review · Area_Chair_zadv · 2024-12-19

**Metareview:**

This paper studies the implicit L_1 regularization (by optimizing a function of the form f(m \odot w) with regularizations on m and w) and discuss how this outperform explicit L_1 regularization. The main idea is that the training dynamics first gives an implicit L_2 regularization and then transitions to L_1 regularization over time. The analysis is based on mirror flow framework. Overall the reviewers think the paper provides interesting new insights on why implicit L_1 regularization works. The main concerns were on the clarity of the paper which has been partly addressed during the response period.

**Additional Comments On Reviewer Discussion:**

The reviewers agreed that the updated version is clearer.

---

### Decision · Program_Chairs · 2025-01-22

Accept (Poster)